# The Target-Charging Technique for Privacy Analysis across Interactive Computations

**Edith Cohen**
Google Research and Tel Aviv University
edith@cohenwang.com

**Xin Lyu**
UC Berkeley
lyuxin1999@gmail.com

## Abstract

We propose the *Target Charging Technique* (TCT), a unified privacy analysis framework for interactive settings where a sensitive dataset is accessed multiple times using differentially private algorithms. Unlike traditional composition, where privacy guarantees deteriorate quickly with the number of accesses, TCT allows computations that don't hit a specified *target*, often the vast majority, to be essentially free (while incurring instead a small overhead on those that do hit their targets). TCT generalizes tools such as the sparse vector technique and top-$k$ selection from private candidates and extends their remarkable privacy enhancement benefits from noisy Lipschitz functions to general private algorithms.

## 1 Introduction

In many practical settings of data analysis and optimization, the dataset $D$ is accessed multiple times interactively via different algorithms $(\mathcal{A}_i)$, so that $\mathcal{A}_i$ depends on the transcript of prior responses $(\mathcal{A}_j(D))_{j<i}$. When each $\mathcal{A}_i$ is privacy-preserving, we are interested in tight end-to-end privacy analysis. We consider the standard statistical framework of differential privacy introduced in [11]. *Composition* theorems [14] are a generic way to do that and achieve overall privacy cost that scales linearly or (via "advanced" composition) with square-root dependence in the number of private computations. We aim for a broad understanding of scenarios where the overall privacy bounds can be lowered significantly via the following paradigm: Each computation is specified by a private algorithm $\mathcal{A}_i$ together with a *target* $\top_i$, that is a subset of its potential outputs. The total privacy cost depends only on computations where the output hits its target, that is $\mathcal{A}_i(D) \in \top_i$. This paradigm is suitable and can be highly beneficial when (i) the specified targets are a good proxy for the actual privacy exposure and (ii) we expect the majority of computations to not hit their target, and thus essentially be "free" in terms of privacy cost.

The Sparse Vector Technique (SVT) [12, 30, 18, 34] is a classic special case. SVT is designed for computations that have the form of approximate threshold tests applied to Lipschitz functions. Concretely, each such `AboveThreshold` test is specified by a 1-Lipschitz function $f$ and a threshold value $t$ and we wish to test whether $f(D) \gtrsim t$. The textbook SVT algorithm compares a noisy value with a noisy threshold (independent Laplace noise for the values and threshold noise that can be updated only after positive responses). Remarkably, the overall privacy cost depends only on positive responses: Roughly, composition is applied to twice the number of positive responses instead of to the total number of computations. In our terminology, the target of each test is a positive response. SVT privacy analysis benefits when the majority of `AboveThreshold` test results are negative (and hence "free"). This makes SVT a key ingredient in a range of methods [13]: private multiplicative weights [18], Propose-Test-Release [10], fine privacy analysis via distance-to-stability [33], model-agnostic private learning [1], and designing streaming algorithms that are robust to adaptive inputs [19, 6].

37th Conference on Neural Information Processing Systems (NeurIPS 2023).

We aim to extend such target-hits privacy analysis to interactive applications of *general* private algorithms (that is, algorithms that provide privacy guarantees but have no other assumptions): private tests, where we would hope to incur privacy cost only for positive responses, and private algorithms that return more complex outputs, e.g., vector average, cluster centers, a sanitized dataset, or a trained ML model, where the goal is to incur privacy cost only when the output satisfies some criteria. Textbook SVT, however, is less amenable to such extensions: First, SVT departs from the natural paradigm of applying private algorithms to the dataset and reporting the output. A natural implementation of private `AboveThreshold` tests would add Laplace noise to the value and compare with the threshold. Instead, SVT takes as input the Lipschitz output of the non-private algorithms with threshold value and the privacy treatment is integrated (added noise both to values and threshold). The overall utility and privacy of the complete interaction are analyzed with respect to the non-private values, which is not suitable when the algorithms are already private. Moreover, the technique of using a hidden shared threshold noise across multiple `AboveThreshold` tests is specific for Lipschitz functions, introduces dependencies between responses, and more critically, results in separate privacy costs for reporting noisy values (that is often required by analytics tasks [22]).

Consider private tests. The natural paradigm is to sequentially choose a test, apply it, and report the result. The hope is to incur privacy loss only on positive responses. Private testing was considered in prior works [21, 7] but in ways that departed from this paradigm: [21] processed the private tests so that a positive answer is returned only when the probability $p$ of a positive response by the private test is very close to 1. This seems unsatisfactory: If the design goal of the private testing algorithm was to report only very high probabilities, then this could have been more efficiently integrated into the design, and if otherwise, then we miss out on acceptable positive responses with moderately high probabilities (e.g. 95%).

Consider now *Top-$k$ selection*, which is a basic subroutine in data analysis, where input algorithms $(\mathcal{A}_i)_{i \in [m]}$ (aka candidates) that return results with quality scores are provided in a *batch* (i.e., non interactively). The selection returns the $k$ candidates with highest quality scores on our dataset. The respective private construct, where the data is sensitive and the algorithms are private, had been intensely studied [23, 17, 32]. The top-$k$ candidates can be viewed as target-hits and we might hope for privacy cost that is close to a composition over $k$ private computations, instead of over $m \gg k$. The natural approach for top-$k$ is *one-shot* (Algorithm 3), where each algorithm is applied once and the responses with top-$k$ scores are reported. Prior works on private selection that achieve this analysis goal include those [9, 29] that use the natural one-shot selection but are tailored to Lipschitz functions (apply the Exponential Mechanism [24] or the Report-Noise-Max paradigm [13]) and works [21, 28, 7] that do apply with general private algorithms but significantly depart from the natural one-shot approach: They make a randomized number of computations that is generally much larger than $m$, with each $\mathcal{A}_i$ invoked multiple times or none. The interpretation of the selection deviates from top-1 and does not naturally extend to top-$k$. We seek privacy analysis that applies to one-shot top-$k$ selection with candidates that are general private algorithms.

The natural interactive paradigm and one-shot selection are simple, interpretable, and general. The departures made in prior works were made for a reason: Simple arguments (that apply with both top-1 one-shot private selection [21] and `AboveThreshold` tests) seem to preclude efficient target-charging privacy analysis: With pure-DP, if we perform $m$ computations that are $\varepsilon$-DP (that is, $m$ candidates or $m$ tests), then the privacy parameter value for a pure DP bound is $\Omega(m)\varepsilon$. With approximate-DP, and even a single "hit," the parameter values are $(\Omega(\varepsilon \log(1/\delta)), \delta)$. The latter suggests a daunting overhead of $O(\log(1/\delta))$ instead of $O(1)$ per "hit." We circumvent the mentioned limitations by taking approximate DP to be a reasonable relaxation and additionally, aim for application regimes where many private computations are performed on the same dataset and we expect multiple, say $\Omega(\log(1/\delta))$, "target hits" (e.g. positive tests and sum of the $k$-values of selections). With these relaxations in place, we seek a unified target-charging analysis (e.g. privacy charge that corresponds to $O(1)$ calls per "target hit") that applies with the natural paradigm across interactive calls and top-$k$ selections.

## 2   Overview of Contributions

We overview our contributions (proofs and details are provided in the Appendix). We introduce the *Target-Charging Technique (TCT)* for privacy analysis over interactive private computations (see Algorithm 1). Each computation performed on the sensitive dataset $D$ is specified by a pair of

private algorithm $\mathcal{A}_i$ and *target* $\top_i$. The interaction is halted after a pre-specified number $\tau$ of computations satisfy $\mathcal{A}_i(D) \in \top_i$. We define targets as follows:

**Definition 2.1** (*q*-Target). *Let $\mathcal{M} : X^n \to \mathcal{Y}$ be a randomized algorithm. For $q \in (0,1]$ and $\varepsilon > 0$, we say that a subset $\top \subseteq \mathcal{Y}$ of outcomes is a $q$-Target of $\mathcal{M}$, if the following holds: For any pair $D^0$ and $D^1$ of neighboring data sets, there exist $p \in [0,1]$, and three distributions $\mathbf{C}$, $\mathbf{B}^0$ and $\mathbf{B}^1$ such that*

1. *The distributions $\mathcal{M}(D^0)$ and $\mathcal{M}(D^1)$ can be written as the following mixtures:*

$$\mathcal{M}(D^0) \equiv p \cdot \mathbf{C} + (1-p) \cdot \mathbf{B}^0,$$
$$\mathcal{M}(D^1) \equiv p \cdot \mathbf{C} + (1-p) \cdot \mathbf{B}^1.$$

2. $\mathbf{B}^0, \mathbf{B}^1$ *are $(\varepsilon, 0)$-indistinguishable,*

3. $\min(\Pr[\mathbf{B}^0 \in \top], \Pr[\mathbf{B}^1 \in \top]) \geq q$.

The effectiveness of a target as a proxy of the actual privacy cost is measured by its $q$-value where $q \in (0,1]$. We interpret $1/q$ as the *overhead factor* of the actual privacy exposure per target hit, that is, the number of private accesses that correspond to a single target hit. Note that an algorithm with a $q$-target for $\varepsilon > 0$ must be $(\varepsilon, 0)$-DP and that any $(\varepsilon, 0)$-DP algorithm has a 1-target, as the set of all outcomes $\top = \mathcal{Y}$ is a 1-target (and hence also a $q$-target for any $q \leq 1$). More helpful targets are "smaller" (so that we are less likely to be charged) with a larger $q$ (so that the overhead per charge is smaller). We establish the following privacy bounds.

**Lemma 2.2** (simplified meta privacy cost of target-charging). *The privacy parameters of Algorithm 1 are $(\varepsilon', \delta)$ where $\varepsilon' \approx \frac{\tau}{q}\varepsilon$ and $\delta = e^{-\Omega(\tau)}$.*

*Alternatively, we obtain parameter values $(\varepsilon', \delta') = (f_\varepsilon(r, \varepsilon), f_\delta(r, \varepsilon) + e^{-\Omega(\tau)})$ where $r \approx \tau/q$ and $(f_\varepsilon(r, \varepsilon), f_\delta(r, \varepsilon))$ are privacy parameter values for advanced composition [14] of $r$ $\varepsilon$-DP computations.*

Proof details for a more general statement that also applies with approximate DP algorithms are provided in Section B (in which case the $\delta$ parameters of all calls simply add up). The idea is simple: We compare the execution of Algorithm 4 on two neighboring data sets $D^0, D^1$. Given a request $(\mathcal{A}, \top)$, let $p, \mathbf{C}, \mathbf{B}, \mathbf{B}^0, \mathbf{B}^1$ be the decomposition of $\mathcal{A}$ w.r.t. $D^0, D^1$ given by Definition 2.1. Then, running $\mathcal{A}$ on $D^0, D^1$ can be implemented in the following equivalent way: we flip a $p$-biased coin. With probability $p$, the algorithm samples from $\mathbf{C}$ and returns the result, without accessing $D^0, D^1$ at all (!). Otherwise, the algorithm needs to sample from $\mathbf{B}^0$ or $\mathbf{B}^1$, depending on whether the private data is $D^0$ or $D^1$. However, by Property 3 in Definition 2.1, there is a probability of at least $q$ that Algorithm 1 will "notice" the privacy-leaking computation by observing a result in the target set $\top$. If this indeed happens, the algorithm increments the counter. On average, each counter increment corresponds to $\frac{1}{q}$ accesses to the private data. Therefore we use the number of target hits (multiplied by $1/q$) as a proxy for the actual privacy leak. Finally, we apply concentration inequalities to obtain high confidence bounds on the probability that the actual number of accesses significantly exceeds its expectation of $\tau/q$. The multiplicative error decreases when the number $\tau$ of target hits is larger. In the regime $\tau > \ln(1/\delta)$, we amortize the mentioned $O(\log(1/\delta))$ overhead of the natural paradigm so that each target hit results in privacy cost equivalent to $O(1/q)$ calls. In the regime of very few target hits (e.g., few private tests or private selections), we still have to effectively "pay" for the larger $\tau = \Omega(\ln(1/\delta))$, but TCT still has some advantages over alternative approaches, due to its use of the natural paradigm and its applicability with general private algorithms.

TCT is simple but turns out to be surprisingly powerful due to natural targets with low overhead. We present an expansive toolkit that is built on top of TCT and describe application scenarios.

## 2.1 `NotPrior` targets

A `NotPrior` target of an $\varepsilon$-DP algorithm is specified by any outcome of our choice (the "prior") that we denote by $\bot$. The `NotPrior` target is the set of all outcomes except $\bot$. Surprisingly perhaps, this is an effective target (See Section C for the proof that applies also with approximate-DP):

**Lemma 2.3** (Property of a `NotPrior` target). *Let $\mathcal{A} : X \to \mathcal{Y} \cup \{\bot\}$, where $\bot \notin \mathcal{Y}$, be an $\varepsilon$-DP algorithm. Then the set of outcomes $\mathcal{Y}$ constitutes an $\frac{1}{e^\varepsilon + 1}$-target for $\mathcal{A}$.*

---

**Algorithm 1:** Target Charging

---

**Input:** Dataset $D = \{x_1, \ldots, x_n\} \in X^n$. Integer $\tau \geq 1$ (Upper limit on the number of target hits).
Fraction $q \in [0, 1]$.

$C \leftarrow 0$                                                                          `// Initialize target hit counter`
**while** $C < \tau$ **do**                                                               `// Main loop`
    **Receive** $(\mathcal{A}, \top)$ where $\mathcal{A}$ is an $\varepsilon$-DP mechanism, and $\top$ is a $q$-target for $\mathcal{A}$
    $r \leftarrow \mathcal{A}(D)$
    **Publish** $r$
    **if** $r \in \top$ **then** $C \leftarrow C + 1$                   `// outcome is a target hit`

---

Note that for small $\varepsilon$, we have $q$ approaching $1/2$ and thus the overhead factor is close to 2. The TCT privacy analysis is beneficial over plain composition when the majority of all outcomes in our interaction match their prior $\bot$. We describe application scenarios for `NotPrior` targets. For most of these scenarios, TCT is the only method we are aware of that provides the stated privacy guarantees in the general context.

**Private testing**   A private test is a private algorithm with a Boolean output. By specifying our prior to be a negative response, we obtain (for small $\varepsilon$) an overhead of 2 for positive responses, which matches SVT. TCT is the only method we are aware of that provides SVT-like guarantees with general private tests.

**Pay-only-for-change**   When we have a prior on the result of each computation and expect the results of most computations to agree with their respective prior, we set $\bot$ to be our prior. We report all results but pay only for those that disagree with the prior. We describe some use cases where paying only for change can be very beneficial (i) the priors are results of the same computations on an older dataset, so they are likely to remain the same (ii) In streaming or dynamic graph algorithms, the input is a sequence of updates where typically the number of changes to the output is much smaller than the number of updates. Differential privacy was used to obtain algorithms that are robust to adaptive inputs [19, 2] by private aggregation of non-robust copies. The pay-only-for-change allows for number of changes to output (instead of the much larger number of updates) that is quadratic in the number of copies. Our result enables such gain with any private aggregation algorithm (that is not necessarily in the form of `AboveThreshold` tests).

## 2.2   Conditional Release

We have a private algorithm $\mathcal{A} : X \rightarrow \mathcal{Y}$ but are interested in the output $\mathcal{A}(D)$ only when a certain condition holds (i.e., when the output is in $\top \subseteq \mathcal{Y}$). The condition may depend on the interaction transcript thus far (depend on prior computations and outputs). We expect most computations not to meet their release conditions and want to be "charged" only for the ones that do. Recall that with differential privacy, not reporting a result also leaks information on the dataset, so this is not straightforward. We define $A_\top := \texttt{ConditionalRelease}(\mathcal{A}, \top)$ as the operation that inputs a dataset $D$, computes $y \leftarrow \mathcal{A}(D)$. If $y \in \top$, then publish $y$ and otherwise publish $\bot$. We show that this operation can be analysed in TCT as a call with the algorithm and `NotPrior` target pair $(\mathcal{A}_\top, \top)$, that is, a target hit occurs if and only if $y \in \top$:

**Lemma 2.4** (`ConditionalRelease` privacy analysis). *$\mathcal{A}_\top$ satisfies the privacy parameters of $\mathcal{A}$ and $\top$ is a `NotPrior` target of $\mathcal{A}_\top$.*

*Proof.* $\mathcal{A}_\top$ processes the output of the private algorithm $\mathcal{A}$ and thus from post processing property is also private with the same privacy parameter values. Now note that $\top$ is a `NotPrior` target of $\mathcal{A}$, with respect to prior $\bot$. $\qquad\square$

We describe some example use-cases:

(i) Private learning of models from the data (clustering, regression, average, ML model) but we are interested in the result only when its quality is sufficient, say above a specified threshold, or when some other conditions hold.

(ii) Greedy coverage or representative selection type applications, where we incur privacy cost only for selected items. To do so, we condition the release on the "coverage" of past responses. For example, when greedily selecting a subset of features that are most relevant or a subset of centers that bring most value.

(iii) Approximate `AboveThreshold` tests on Lipschitz functions, with release of above-threshold noisy values: As mentioned, SVT incurs additional privacy cost for the reporting whereas TCT (using `ConditionalRelease`) does not, so TCT benefits in the regime of sufficiently many target hits.

(iv) `AboveThreshold` tests with sketch-based approximate distinct counts: Distinct counting sketches [16, 15, 5] meet the privacy requirement by the built-in sketch randomness [31]. We apply `ConditionalRelease` and set $\top$ to be above threshold values. In comparison, despite the function (distinct count) being 1-Lipschitz, the use of SVT for this task incurs higher overheads in utility (approximation quality) and privacy: Even for the goal of just testing, a direct use of SVT treats the approximate value as the non-private input, which reduces accuracy due to the additional added noise. Treating the reported value as a noisy Lipschitz still incurs accuracy loss due to the threshold noise, threshold noise introduces bias, and analysis is complicated by the response not following a particular noise distribution. For releasing values, SVT as a separate distinct-count sketch is needed to obtain an independent noisy value [22], which increases both storage and privacy costs.

## 2.3 Conditional Release with Revisions

We present an extension of Conditional Release that allows for followup *revisions* of the target. The initial `ConditionalRelease` and the followup `ReviseCR` calls are described in Algorithm 2. The `ConditionalRelease` call specifies a computation identifier $h$ for later reference, an algorithm and a target pair $(\mathcal{A}, \top)$. It draws $r_h \sim \mathcal{A}(D)$ and internally stores $r_h$ and a current target $\top_h \leftarrow \top$. When $r_h \in \top$ then $r_h$ is published and a charge is made. Otherwise, $\bot$ is published. Each (followup) `ReviseCR` call specifies an identifier $h$ and a disjoint extension $\top'$ to its current target $\top_h$. If $r_h \in \top'$, then $r_h$ is published and a charge is made. Otherwise, $\bot$ is published. The stored current target for computation $h$ is augmented to include $\top'$. Note that a target hit occurs at most once in a sequence of (initial and followup revise) calls and if and only if the result of the initial computation $r_h$ is in the final target $\top_h$.

---

**Algorithm 2:** Conditional Release and Revise Calls

```
// Initial Conditional Release call: Analysed in TCT as a (ε,δ)-DP algorithm 𝒜_⊤ and NotPrior target
   ⊤
Function ConditionalRelease(h, 𝒜, ⊤):      // unique identifier h, an (ε,δ)-DP algorithm 𝒜 → 𝒴,
   ⊤ ⊂ 𝒴
   | ⊤_h ← ⊤                                        // Current target for computation h
   | TCT Charge for δ                                  // If δ > 0, see Section B
   | r_h ← 𝒜(D)                                        // Result for computation h
   | if r_h ∈ ⊤_h then                     // publish and charge only if outcome is in ⊤_h
   |   | Publish r_h
   |   | TCT Charge for a NotPrior target hit of an ε-DP algorithm
   | else
   |   | Publish ⊥

// Revise call: Analysed in TCT as a 2ε-DP Algorithm (𝒜 | ¬⊤_h)_⊤' and NotPrior target ⊤'
Function ReviseCR(h, ⊤'):                              // Revise target to include ⊤'
   | Input: An identifier h of a prior ConditionalRelease call, target extension ⊤' where ⊤' ∩ ⊤_h = ∅
   | if r_h ∈ ⊤' then                          // Result is in current target, publish and charge
   |   | Publish r_h
   |   | TCT Charge for a NotPrior target hit of an 2ε-DP algorithm
   | else
   |   | Publish ⊥
   | ⊤_h ← ⊤_h ∪ ⊤'                              // Update the target to include extension
```

---

We show the following (Proof provided in Section D):

**Lemma 2.5** (Privacy analysis for Algorithm 2). *Each `ReviseCR` call can be analysed in TCT as a call to a $2\varepsilon$-DP algorithm with a `NotPrior` target $\top'$.*

Thus, the privacy cost of conditional release followed by a sequence of revise calls is within a factor of 2 (due to the doubled privacy parameter on revise calls) of a single `ConditionalRelease` call made with the final target. The revisions extension of conditional release facilitates our results for private selection, which are highlighted next.

### 2.4 Private Top-$k$ Selection

Consider the nature one-shot top-$k$ selection procedure as shown in Algorithm 3: We call each algorithm once and report the $k$ responses with the highest quality scores. We establish the following:

**Lemma 2.6** (Privacy of One-Shot Top-$k$ Selection). *Consider one-shot top-$k$ selection (Algorithm 3) on a dataset $D$ where $\{\mathcal{A}_i\}$ are $(\varepsilon, \delta_i)$-DP. This selection can be simulated exactly in TCT by a sequence of calls to $(2\varepsilon, \delta)$-DP algorithms with `NotPrior` targets that has $k$ target hits.*

*As a corollary, assuming $\varepsilon < 1$, Algorithm 3 is $(O(\varepsilon\sqrt{k\log(1/\delta)}), 2^{-\Omega(k)} + \delta + \sum_i \delta_i)$-DP for every $\delta > 0$.*

To the best of our knowledge, our result is the first such bound for one-shot selection from general private candidates. For the case when the only computation performed on $D$ is a single top-1 selection, we match the "bad example" in [21] (see Theorem J.1). In the regime where $k > \log(1/\delta)$ our bounds generalize those specific to Lipschitz functions in [9, 29] (see Section J). Moreover, Lemma 2.6 allows for a unified privacy analysis of interactive computations that are interleaved with one-shot selections. We obtain $O(1)$ overhead per target hit when there are $\Omega(\log(1/\delta))$ hits in total.

---

**Algorithm 3:** One-Shot Top-$k$ Selection

**Input:** A dataset $D$. Candidate algorithms $\mathcal{A}_1, \ldots, \mathcal{A}_m$. Parameter $k \leq m$.
$S \leftarrow \emptyset$
**for** $i = 1, \ldots, m$ **do**
    $(y_i, s_i) \leftarrow \mathcal{A}_i(D)$
    $S \leftarrow S \cup \{(i, y_i, s_i)\}$
**return** $L \leftarrow$ *the top-$k$ triplets from $S$, by decreasing $s_i$*

---

The proofs of Lemma 2.6 and implications to selection tasks are provided in Section J. The proof utilizes Conditional Release with revisions (Section 2.3).

#### 2.4.1 Selection using Conditional Release

We analyze private selection procedures using conditional release (see Section J for details). First note that `ConditionalRelease` calls (without revising) suffice for *one-shot above-threshold* selection (release all results with a quality score that exceeds a pre-specified threshold $t$), with target hits only on what was released: We simply specify the release condition to be $s_i > t$. What is missing in order to implement one-shot top-$k$ selection is an ability to find the "right" threshold (a value $t$ so that exactly $k$ candidates have quality scores above $t$), while incurring only $k$ target hits. The revise calls provide the functionality of lowering the threshold of previous conditional release calls (lowering the threshold amounts to augmenting the target). This functionality allows us to simulate a sweep of the $m$ results of the batch in the order of decreasing quality scores. We can stop the sweep when a certain condition is met (the condition must be based on the prefix of the ordered sequence that we viewed so far) and we incur target hits only for the prefix. To simulate a sweep, we run a high threshold conditional release of all $m$ candidates and then incrementally lower the threshold using sets of $m$ revise calls (one call per candidate). The released results are in decreasing order of quality scores. To prove Lemma 2.6 we observe that the one-shot top-$k$ selection (Algorithm 3) is simulated exactly by such a sweep that halts after $k$ scores are released (the sweep is only used for analysis).

As mentioned, with this approach we can apply *any stopping condition that depends on the prefix*. This allows us to use data-dependent selection criteria. One natural such criteria (instead of using

a rigid value of $k$) is to choose $k$ when there is a large gap in the quality scores, that the $(k+1)$st quality score is much lower than the $k$th score [35]. This criterion can be implemented using a one-shot algorithm and analyzed in the same way using an equivalent sweep. Data-dependent criteria are also commonly used in applications such as clustering (choose "the right" number of clusters according to gap in clustering cost) and greedy selection of representatives.

### 2.5   Best of multiple targets

*Multi-target* charging is a simple but useful extension of Algorithm 1 (that is "single target"). With $k$-TCT, queries have the form $\big(\mathcal{A}, (\top_i)_{i \in [k]}\big)$ where $\top_i$ for $i \in [k]$ are $q$-targets (we allow targets to overlap). The algorithm maintains $k$ counters $(C_i)_{i \in [k]}$. For each query, for each $i$, we increment $C_i$ if $r \in \top_i$. We halt when $\min_i C_i = \tau$.

The multi-target extension allows us to flexibly reduce the total privacy cost to that of the "best" among $k$ target indices *in retrospect* (the one that is hit the least number of times). Interestingly, this extension is almost free in terms of privacy cost: The number of targets $k$ only multiplies the $\delta$ privacy parameter (see Section B.1 for details).

### 2.6   `BetweenThresholds` in TCT

The `BetweenThresholds` classifier refines the `AboveThreshold` test: The goal is to report if the noisy Lipschitz value is `below`, `between`, or `above` two thresholds $t_l < t_r$. We aim for privacy loss that only depends on `between` outcomes. An SVT-based `BetweenThresholds` was proposed by [3] (with noisy value and thresholds). Their analysis required the gap size to satisfy $t_r - t_l \geq (12/\varepsilon)(\log(10/\varepsilon) + \log(1/\delta) + 1)$.

We consider the "natural" `BetweenThresholds` classifier that compares the Lipschitz value with added $\mathbf{Lap}(1/\varepsilon)$ noise to the two threshold values and reports the result. This is $\varepsilon$-DP and we show (see Section G) that the `between` outcome is a target with $q \geq (1 - e^{-(t_r - t_l)\varepsilon}) \cdot \frac{1}{e^\varepsilon + 1}$. This $q$ value is at most that of a `NotPrior` target ($\frac{1}{e^\varepsilon + 1}$) and degrades smoothly as the gap decreases. Importantly, there is almost no degradation for fairly small gaps: When $t_r - t_l \geq 2/\varepsilon$ (resp. $3/\epsilon$) the $q$-value is $\approx \frac{0.87}{e^\varepsilon + 1}$ (resp $\frac{0.95}{e^\varepsilon + 1}$).

One benefit of TCT `BetweenThresholds` is that it applies with much smaller gaps $t_r - t_l$ compared with [3], also asymptotically. Another benefit that holds even for large gaps (where the SVT variant is applicable) is that the natural algorithm requires lower privacy noise for a given accuracy. In Section H we demonstrate this improvement numerically in that we can answer $\times 6$-$\times 95$ fold more queries for the same accuracy requirements compared to [3]. This bring `BetweenThresholds` into the practical regime.

We can compare an `AboveThreshold` test with a threshold $t$ with a `BetweenThresholds` classifier with $t_l = t - 1/\varepsilon$ and $t_r = t + 1/\varepsilon$. Surprisingly perhaps, despite `BetweenThresholds` being *more informative* than `AboveThreshold`, as it provides more granular information on the value, its privacy cost is *lower*, and is much lower when values are either well above or well below the thresholds. Somehow, the addition of the third `between` outcome to the test tightened the privacy analysis! A natural question is whether we can extend this benefit more generally – inject a "boundary outcome" as a target when our private algorithm does not have one, then tighten the privacy analysis when queries are "far" from the boundary. We introduce next a method that achieves this goal.

### 2.7   The Boundary Wrapper Method

When the algorithm is a tester or a classifier, the result is most meaningful when one outcome dominates the distribution $\mathcal{A}(D)$. Moreover, when performing a sequence of tests or classification tasks we might expect most queries to have high confidence labels (e.g., [27, 1]). Our hope then is to incur privacy cost that depends only on the "uncertainty," captured by the probability of non-dominant outcomes. When we have for each computation a good prior on which outcome is most likely, this goal can be achieved via `NotPrior` targets (Section 2.1). When we expect the whole sequence to be dominated by one type of outcome, even when we don't know which one it is, this goal can be achieved via `NotPrior` with multiple targets (Section 2.5). But these approaches do not apply when a dominant outcome exists in most computations but we have no handle on it.

For a private test $\mathcal{A}$, can we choose a moving target *per computation* to be the value with the smaller probability $\arg\min_{b\in\{0,1\}} \Pr[\mathcal{A}(D) = b]$? More generally, with a private classifier, can we somehow choose the target to be all outcomes except for the most likely one? Our *boundary wrapper*, described in Algorithm 4, achieves that goal. The privacy wrapper $\mathcal{W}$ takes any private algorithm $\mathcal{A}$, such as a tester or a classifier, and wraps it to obtain algorithm $\mathcal{W}(\mathcal{A})$. The wrapped algorithm has its outcome set augmented to include one *boundary* outcome $\top$ that is designed to be a $q$-target. The wrapper returns $\top$ with some probability that depends on the distribution of $\mathcal{A}(D)$ and otherwise returns a sample from $\mathcal{A}(D)$ (that is, the output we would get when directly applying $\mathcal{A}$ to $D$). We then analyse the wrapped algorithm in TCT.

Note that the probability of the wrapper $\mathcal{A}$ returning $\top$ is at most $1/3$ and is roughly proportional to the probability of sampling an outcome other than the most likely from $\mathcal{A}(D)$. When there is no dominant outcome the $\top$ probability tops at $1/3$. Also note that a dominant outcome (has probability $p \in [1/2, 1]$ in $\mathcal{A}(D)$) has probability $p/(2-p)$ to be reported. This is at least $1/3$ when $p = 1/2$ and is close to 1 when $p$ is close to 1. For the special case of $\mathcal{A}$ being a private test, there is always a dominant outcome.

A wrapped `AboveThreshold` test provides the benefit of `BetweenThresholds` discussed in Section 2.6 where we do not pay privacy cost for values that are far from the threshold (on either side). This is achieved mechanically without the need to explicitly introduce two thresholds around the given one and defining a different algorithm.

---

**Algorithm 4:** Boundary Wrapper

**Input:** Dataset $D = \{x_1, \ldots, x_n\} \in X^n$, a private algorithm $\mathcal{A}$

$r^* \leftarrow \arg\max_r \Pr[\mathcal{A}(D) = r]$         // The most likely outcome of $\mathcal{A}(D)$

$\pi(D) \leftarrow 1 - \Pr[\mathcal{A}(D) = r^*]$     // Probability that $\mathcal{A}$ does not return the most likely outcome

$c \sim \mathbf{Ber}(\min\left\{\frac{1}{3}, \frac{\pi}{1+\pi}\right\})$           // Coin toss for boundary

**if** $c = 1$ **then Return** $\top$ **else Return** $\mathcal{A}(D)$        // return boundary or value

---

We show (proofs provided in Section E) that the wrapped algorithm is nearly as private as its baseline:

**Lemma 2.7** (Privacy of a wrapped algorithm)**.** *If $\mathcal{A}$ is $\varepsilon$-DP then Algorithm 4 applied to $\mathcal{A}$ is $t(\varepsilon)$-DP where $t(\varepsilon) \leq \frac{4}{3}\varepsilon$.*

**Lemma 2.8** ($q$-value of the boundary target)**.** *The outcome $\top$ of a boundary wrapper (Algorithm 4) of an $\varepsilon$-DP algorithm is a $\frac{e^{t(\varepsilon)}-1}{2(e^{\varepsilon+t(\varepsilon)}-1)}$-target.*

For small $\varepsilon$ we obtain $q \approx t(\varepsilon)/(2(\varepsilon + t(\varepsilon)))$. Substituting $t(\varepsilon) = \frac{4}{3}\varepsilon$ we obtain $q \approx \frac{2}{7}$. Since the target $\top$ has probability at most $1/3$, this is a small loss of efficiency ($1/6$ factor overhead) compared with composition in the worst case when there are no dominant outcomes.

The boundary wrapper yields light-weight privacy analysis that pays only for the "uncertainty" of the response distribution $\mathcal{A}(D)$ and can be an alternative to more complex approaches based on smooth sensitivity (the stability of $\mathcal{A}(D)$ to changes in $D$) [25, 10, 33]. Note that the boundary-wrapper method assumes availability of the probability of the most dominant outcome in the distribution $\mathcal{A}(D)$, when it is large enough. The probability can always be computed without incurring privacy costs (only computation cost) and is readily available with the Exponential Mechanism [24] or when applying known noise distributions for `AboveThreshold`, `BetweenThresholds`, and Report-Noise-Max [9]. In Section F we propose a boundary-wrapper that only uses sampling access to $\mathcal{A}(D)$.

### 2.7.1 Applications to Private Learning using Non-privacy-preserving Models

Methods that achieve private learning through training non-private models include Private Aggregation of Teacher Ensembles (PATE) [26, 27] and Model-Agnostic private learning [1]. The private dataset $D$ is partitioned into $k$ parts $D = D_1 \sqcup \cdots \sqcup D_k$ and a model is trained (non-privately) on each part. For multi-class classification with $c$ labels, the trained models can be viewed as functions $\{f_i : \mathcal{X} \to [c]\}_{i\in[k]}$. Note that changing one sample in $D$ can only change the training set of one of the models. To privately label an example $x$ drawn from a public distribution, we compute the

predictions of all the models $\{f_i(x)\}_{i\in[k]}$ and consider the counts $n_j = \sum_{i\in[k]} \mathbf{1}\{f_i(x) = j\}$ (the number of models that gave label $j$ to example $x$) for $j \in [c]$. We then privately aggregate to obtain a private label, for example using the Exponential Mechanism [24] or Report-Noisy-Max [9, 29]. This setup is used to process queries (label examples) until the privacy budget is exceeded. In PATE, the new privately-labeled examples are used to train a new *student* model (and $\{f_i\}$ are called *teacher* models). In these applications we seek tight privacy analysis. Composition over all queries – for $O(1)$ privacy, only allows for $O(k^2)$ queries. We aim to replace this with $O(k^2)$ "target hits." These works used a combination of methods including SVT, smooth sensitivity, distance-to-instability, and propose-test-release [10, 33]. The TCT toolkit can streamline the analysis:

(i) It was noted in [1, 27] that when the teacher models are sufficiently accurate, we can expect that $n_j \gg k/2$ on the ground truth label $j$ on most queries. High-agreement examples are also more useful for training the student model. Moreover, agreement implies stability and lower privacy cost (when accounted through the mentioned methods) is lower. Instead, to gain from this stability, we can apply the boundary wrapper (Algorithm 4) on top of the Exponential Mechanism. Then use $\top$ as our target. Agreement queries, where $\max_j n_j \gg k/2$ (or more finely, when $h = \arg\max_j n_j$ and $n_h \gg \max_{j\in[k]\setminus\{h\}} n_j$) are very unlikely to be target hits.

(ii) If we expect most queries to be either high agreement $\max_j n_j \gg k/2$ or no agreement $\max_j n_j \ll k/2$ and would like to avoid privacy charges also with no agreement, we can apply `AboveThreshold` test to $\max_j n_j$. If above, we apply the exponential mechanism. Otherwise, we report "Low." The wrapper applied to the combined algorithm returns a label in $[c]$, "Low," or $\top$. Note that "Low" is a dominant outcome with no-agreement queries (where the actual label is not useful anyway) and a class label in $[c]$ is a dominant outcome with high agreement. We therefore incur privacy loss only on weak agreements.

## 2.8 SVT with individual privacy charging

Our TCT privacy analysis simplifies and improves the analysis of SVT with individual privacy charging, introduced by Kaplan et al [20]. The input is a dataset $D \in \mathcal{X}^n$ and an online sequence of linear queries that are specified by predicate and threshold value pairs $(f_i, T_i)$. For each query, the algorithms reports noisy `AboveThreshold` test results $\sum_{x\in D} f_i(x) \gtrsim T$. Compared with the standard SVT, which halts after reporting $\tau$ positive responses, SVT with individual charging maintains a separate budget counter $C_x$ for each item $x$. For each query with a positive response, the algorithm only charges items that *contribute* to this query (namely, all the $x$'s such that $f_i(x) = 1$). Once an item $x$ contributes to $\tau$ hits (that is, $C_x = \tau$), it is removed from the data set. This finer privacy charging improves utility with the same privacy budget, as demonstrated by several recent works [20, 6]. We establish the following:

**Theorem 2.9** (Privacy of Algorithm 5). *Assume $\varepsilon < 1$. Algorithm 5 is $(O(\sqrt{\tau \log(1/\delta)}\varepsilon), 2^{-\Omega(\tau)} + \delta)$-DP for every $\delta \in (0, 1)$.*

Compared with prior work [20]: Algorithm 5 simply adds Laplace noise to obtain $\hat{f}_i = \left(\sum_{x\in D} f_i(x)\right) + \mathbf{Lap}(1/\varepsilon)$ and then tests whether $\hat{f}_i \geq T$, whereas [20] adds two independent Laplace noises and publishes the approximate sum $\hat{f}_i$ for "Above-Threshold" without incurring additional privacy loss. Our analysis is much simpler (few lines instead of several pages) and significantly tighter, also asymptotically: For the privacy bound in the statement of Theorem 5, our additive error is lower by a $\log(1/\varepsilon)\sqrt{\log(1/\delta)}$ factor. Importantly, our improvement aligns the bounds of SVT with individual privacy charging with those of standard SVT, bringing the former into the practical regime. See Section I for details.

**Conclusion**  We introduced the Target Charging Technique (TCT), a versatile unified privacy analysis framework that is particularly suitable when a sensitive dataset is accessed multiple times via differentially private algorithms. We provide an expansive toolkit and demonstrate significant improvement over prior work for basic tasks such as private testing and one-shot selection, describe use cases, and list challenges for followup works. TCT is simple with low overhead and we hope will be adopted in practice.

---

**Algorithm 5:** SVT with Individual Privacy Charging

---

**Input:** Private data set $D \in \mathcal{X}^n$; privacy budget $\tau > 0$; Privacy parameter $\varepsilon > 0$.

**foreach** $x \in D$ **do** $C_x \leftarrow 0$        `// Initialize a counter for item x`

**for** $i = 1, 2, \ldots,$ **do**        `// Receive queries`

    **Receive** a predicate $f_i : \mathcal{X} \to [0, 1]$ and threshold $T_i \in \mathbb{R}$

    $\hat{f}_i \leftarrow \left( \sum_{x \in D} f_i(x) \right) + \mathbf{Lap}(1/\varepsilon)$        `// Add Laplace noise to count`

    **if** $\hat{f}_i \geq T_i$ **then**        `// Compare with threshold`

        **Publish** $\hat{f}_i$

        **foreach** $x \in D$ *such that* $f(x) > 0$ **do**

            $C_x \leftarrow C_x + 1$

            **if** $C_x = \tau$ **then** Remove $x$ from $D$

    **else** **Publish** $\perp$

---

**Acknowledgement**   Edith Cohen is partially supported by Israel Science Foundation (grant no. 1156/23)

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

# A Preliminaries

**Notation.** We say that a function $f$ over datasets is $t$-Lipschitz if for any two neighboring datasest $D^0$, $D^1$, it holds that $|f(D^1) - f(D^0)| \leq t$. For two reals $a, b \geq 0$ and $\varepsilon > 0$, we write $a \approx_\varepsilon b$ if $e^{-\varepsilon} b \leq a \leq e^\varepsilon b$.

For two random variables $X^0, X^1$, we say that they are $\varepsilon$-indistinguishable, denoted $X^0 \approx_\varepsilon X^1$, if their max-divergence and symmetric counterpart are both at most $\varepsilon$. That is, for $b \in \{0, 1\}$, $\max_{S \subseteq \mathsf{supp}(X^b)} \ln \left[ \frac{\Pr[X^b \in S]}{\Pr[X^{1-b} \in S]} \right] \leq \varepsilon$.

We similarly say that for $\delta > 0$, the random variables are $(\varepsilon, \delta)$-indistinguishable, denoted $X^0 \approx_{\varepsilon, \delta} X^1$, if for $b \in \{0, 1\}$

$$\max_{S \subseteq \mathsf{supp}(X^b)} \ln \left[ \frac{\Pr[X^b \in S] - \delta}{\Pr[X^{1-b} \in S]} \right] \leq \varepsilon.$$

For two probability distributions, $\mathcal{B}^0, \mathcal{B}^1$ We extend the same notation and write $\mathbf{B}^0 \approx_\varepsilon \mathbf{B}^1$ and $\mathbf{B}^0 \approx_{\varepsilon, \delta} \mathbf{B}^1$ when this holds for random variables drawn from the respective distributions.

The following relates $(\varepsilon, 0)$ and $(\varepsilon, \delta)$-indistinguishability with $\delta = 0$ and $\delta > 0$.

**Lemma A.1.** *Let $\mathbf{B}^0$, $\mathbf{B}^1$ be two distributions. Then $\mathbf{B}^0 \approx_{\varepsilon, \delta} \mathbf{B}^1$ if and only if we can express them as mixtures*

$$\mathbf{B}^b \equiv (1 - \delta) \cdot \mathbf{N}^b + \delta \cdot \mathbf{E}^b \,,$$

*where $\mathbf{N}^0 \approx_\varepsilon \mathbf{N}^1$.*

We treat random variables interchangeably as distributions, and in particular, for a randomized algorithms $\mathcal{A}$ and input $D$ we use $\mathcal{A}(D)$ to denote both the random variable and the distribution.

For a space of datasets equipped with a symmetric *neighboring* relation, we say an algorithm $\mathcal{A}$ is $\varepsilon$-DP (pure differential privacy), if for any two neighboring datasets $D$ and $D'$, $\mathcal{A}(D) \approx_\varepsilon \mathcal{A}(D')$. Similarly, we say $\mathcal{A}$ is $(\varepsilon, \delta)$-DP (approximate differential privacy) if for any two neighboring datasets $D, D'$, it holds that $\mathcal{A}(D) \approx_{\varepsilon, \delta} \mathcal{A}(D')$ [11]. We refer to $\varepsilon, \delta$ as the *privacy parameters*.

A *private test* is a differentially private algorithm with Boolean output (say in $\{0, 1\}$).

**Remark A.2.** *The literature in differential privacy uses different definitions of neighboring datasets. TCT, and properties in these preliminaries, apply with any symmetric relation over the space of possible input datasets.*

The following is immediate from Lemma A.1:

**Corollary A.3** (Decomposition of an approximate DP Algorithm)**.** *An algorithm $\mathcal{A}$ is $(\varepsilon, \delta)$-DP if and only if for any two neighboring datasets $D^0$ and $D^1$ we can represent each distribution $\mathcal{A}(D^b)$ ($b \in \{0, 1\}$) as a mixture*

$$\mathcal{A}(D^b) \equiv (1 - \delta) \cdot \mathbf{N}^b + \delta \cdot \mathbf{E}^b \,,$$

*where $\mathbf{N}^0 \approx_\varepsilon \mathbf{N}^1$.*

Differential privacy satisfies the post-processing property (post-processing of the output of a private algorithm remains private with the same parameter values) and also has nice composition theorems:

**Lemma A.4** (DP composition [11, 14])**.** *An interactive sequence of $r$ executions of $\varepsilon$-DP algorithms satisfies $(\varepsilon', \delta)$-DP for*

- $\varepsilon' = r\varepsilon$ *and $\delta = 0$ by basic composition* [11]*, or*

- *for any $\delta > 0$,*

$$\varepsilon' = \frac{1}{2} r \varepsilon^2 + \varepsilon \sqrt{2r \log(1/\delta)} \,.$$

*by advanced composition* [14]*.*

## A.1 Simulation-based privacy analysis

Privacy analysis of an algorithm $\mathcal{A}$ via simulations is performed by simulating the original algorithm $\mathcal{A}$ on two neighboring datasets $D^0, D^1$. The simulator does not know which of the datasets is the actual input (but knows everything about the datasets). Another entity called the "data holder" has the 1-bit information $b \in \{0, 1\}$ on which dataset it is. We perform privacy analysis with respect to what the holder discloses to the simulator regarding the private bit $b$ (taking the maximum over all choices of $D^0, D^1$). The privacy analysis is worst case over the choices of two neighboring datasets. This is equivalent to performing privacy analysis for $\mathcal{A}$.

**Lemma A.5** (Simulation-based privacy analysis). *[8] Let $\mathcal{A}$ be an algorithm whose input is a dataset. If there exist a pair of interactive algorithms $\mathcal{S}$ and $H$ satisfying the following 2 properties, then algorithm $\mathcal{A}$ is $(\varepsilon, \delta)$-DP.*

1. *For every two neighboring datasets $D^0, D^1$ and for every bit $b \in \{0, 1\}$ it holds that*

$$\big(\mathcal{S}(D^0, D^1) \leftrightarrow H(D^0, D^1, b)\big) \equiv \mathcal{A}(D^b).$$

   *Here $\big(\mathcal{S}(D^0, D^1) \leftrightarrow H(D^0, D^1, b)\big)$ denotes the outcome of $\mathcal{S}$ after interacting with $H$.*

2. *Algorithm $H$ is $(\varepsilon, \delta)$-DP w.r.t. the input bit $b$.*

## A.2 Privacy Analysis with Failure Events

Privacy analysis of a randomized algorithm $\mathcal{A}$ using designated failure events is as follows:

1. Designate some runs of the algorithm as *failure events*.
2. Compute an upper bound on the maximum probability, over datasets $D$, of a transcript with a failure designation.
3. Analyse the privacy of the interaction transcript conditioned on no failure designation.

Note that the failure designation is only used for the purpose of analysis. The output on failure runs is not restricted (e.g., could be the dataset $D$)

**Lemma A.6** (Privacy analysis with privacy failure events). *Consider privacy analysis of $\mathcal{A}$ with failure events. If the probability of a failure event is bounded by $\delta^* \in [0, 1]$ and the transcript conditioned on non-failure is $(\varepsilon', \delta')$-DP then the algorithm $\mathcal{A}$ is $(\varepsilon, \delta + \delta^*)$-DP.*

*Proof.* Let $D^0$ and $D^1$ be neighboring datasets. From our assumptions, for $b \in \{0, 1\}$, we can represent $\mathcal{A}(D^b)$ as the mixture $\mathcal{A}(D^b) \equiv (1 - \delta^b) \cdot \mathbf{Z}^b + \delta^b \cdot \mathbf{F}^b$, where $\mathbf{Z}^0 \approx_{\varepsilon', \delta'} \mathbf{Z}^1$, and $\delta^{(b)} \leq \delta^*$. From Lemma A.1, we have $\mathbf{Z}^b \equiv (1 - \delta') \cdot \mathbf{N}^b + \delta' \cdot \mathbf{E}^b$, where $\mathbf{N}^0 \approx_{\varepsilon'} \mathbf{N}^1$.

Then

$$
\begin{aligned}
\mathcal{A}(D^b) &= (1 - \delta^{(b)}) \cdot \mathbf{Z}^b + \delta^{(b)} \cdot \mathbf{F}^{(b)} \\
&= (1 - \delta^*) \cdot \mathbf{Z}^b + (\delta^* - \delta^{(b)}) \cdot \mathbf{Z}^b + \delta^{(b)} \cdot \mathbf{F}^b \\
&= (1 - \delta^*) \cdot \mathbf{Z}^b + \delta^* \cdot \Big((1 - \delta^{(b)}/\delta^*) \cdot \mathbf{Z}^b + \delta^{(b)} \cdot \mathbf{F}^b\Big) \\
&= (1 - \delta^*)(1 - \delta') \cdot \mathbf{N}^b + (1 - \delta^*)\delta' \cdot \mathbf{E}^b + \delta^* \cdot \Big((1 - \delta^{(b)}/\delta^*) \cdot \mathbf{Z}^b + \delta^{(b)} \cdot \mathbf{F}^b\Big) \\
&= (1 - \delta^* - \delta') \cdot \mathbf{N}^b + \delta'\delta^* \cdot \mathbf{N} + (1 - \delta^*)\delta' \cdot \mathbf{E}^b + \delta^* \cdot \Big((1 - \delta^{(b)}/\delta^*) \cdot \mathbf{Z}^b + \delta^{(b)} \cdot \mathbf{F}^b\Big)
\end{aligned}
$$

The claim follows from Corollary A.3. $\qquad\qquad\square$

Using simulation-based privacy analysis we can treat an interactive sequence of approximate-DP algorithms (optionally with designated failure events) as a respective interactive sequence of pure-DP algorithms where the $\delta$ parameters are anlaysed through failure events. This simplifies analysis:

We can relate the privacy of a composition of approximate-DP algorithms to that of a composition of corresponding pure-DP algorithms:

**Corollary A.7** (Composition of approximate-DP algorithms)**.** *An interactive sequence of $(\varepsilon_i, \delta_i)$-DP algorithms ($i \in [k]$) has privacy parameter values $(\varepsilon', \delta' + \sum_{i=1}^k \delta_i)$, where $(\varepsilon', \delta')$ are privacy parameter values of a composition of pure $(\varepsilon_i, 0)$-DP algorithms $i \in [k]$.*

*Proof.* We perform simulation-based analysis. Fix two neighboring datasets $D^0, D^1$. For an $(\varepsilon_i, \delta_i)$-DP algorithm, we can consider the mixtures as in Corollary A.3. We draw $c \sim \mathbf{Ber}(\delta_i)$ and if $c = 1$ designate the output as failure and return $r \sim \mathbf{E}^{(b)}$. Otherwise, we return $r \sim \mathbf{N}^{(b)}$. The overall failure probability is bounded by $1 - \prod_i (1 - \delta_i) \leq \sum_i \delta_i$. The output conditioned on non-failure is a composition of $(\varepsilon_i, 0)$-DP algorithms ($i \in [k]$). The claim follows using Lemma A.6. $\square$

# B   The Target-Charging Technique

We extend the definition of $q$-targets (Definition 2.1) so that it applies with approximate DP algorithms:

**Definition B.1** ($q$-target with $(\varepsilon, \delta)$ of a pair of distributions)**.** *Let $\mathcal{A} \to \mathcal{Y}$ be a randomized algorithm. Let $\mathbf{Z}^0$ and $\mathbf{Z}^1$ be two distributions with support $\mathcal{Y}$. We say that $\top \subseteq \mathcal{Y}$ is a $q$-target of $(\mathbf{Z}^0, \mathbf{Z}^1)$ with $(\varepsilon, \delta)$, where $\varepsilon > 0$ and $\delta \in [0, 1)$, if there exist $p \in [0, 1]$ and five distributions $\mathbf{C}$, $\mathbf{B}^b$, and $\mathbf{E}^b$ (for $b \in \{0, 1\}$) such that $\mathbf{Z}^0$ and $\mathbf{Z}^1$ can be written as the mixtures*

$$\mathbf{Z}^0 \equiv (1 - \delta) \cdot (p \cdot \mathbf{C} + (1 - p) \cdot \mathbf{B}^0) + \delta \cdot \mathbf{E}^0$$
$$\mathbf{Z}^1 \equiv (1 - \delta) \cdot (p \cdot \mathbf{C} + (1 - p) \cdot \mathbf{B}^1) + \delta \cdot \mathbf{E}^1$$

*where $\mathbf{B}^0 \approx_\varepsilon \mathbf{B}^1$, and $\min(\Pr[\mathbf{B}^0 \in \top], \Pr[\mathbf{B}^1 \in \top]) \geq q$.*

**Definition B.2** ($q$-target with $(\varepsilon, \delta)$ of a randomized algorithm)**.** *Let $\mathcal{A} \to \mathcal{Y}$ be a randomized algorithm. We say that $\top \subseteq \mathcal{Y}$ is a $q$-target of $\mathcal{A}$ with $(\varepsilon, \delta)$, where $\varepsilon > 0$ and $\delta \in [0, 1)$, if for any pair $D^0, D^1$ of neighboring datasets, $\top$ is a $q$-target with $(\varepsilon, \delta)$ of $\mathcal{A}(D^0)$ and $\mathcal{A}(D^1)$.*

We can relate privacy of an algorithms or indistinguishability of two distributions to existence of $q$-targets:

**Lemma B.3.** *(i) If $(\mathbf{Z}^0, \mathbf{Z}^1)$ have a $q$-target with $(\varepsilon, \delta)$ then $\mathbf{Z}^0 \approx_{\varepsilon, \delta} \mathbf{Z}^1$. Conversely, if $\mathbf{Z}^0 \approx_{\varepsilon, \delta} \mathbf{Z}^1$ then $(\mathbf{Z}^0, \mathbf{Z}^1)$ have a 1-target with $(\varepsilon, \delta)$ (the full support is a 1-target).*

*(ii) If an algorithm $\mathcal{A}$ has a $q$-target with $(\varepsilon, \delta)$ then $\mathcal{A}$ is $(\varepsilon, \delta)$-DP. Conversely, if an algorithm $\mathcal{A}$ is $(\varepsilon, \delta)$-DP then it has a 1-target (the set $\mathcal{Y}$) with $(\varepsilon, \delta)$.*

*Proof.* If two distributions $\mathbf{B}^0, \mathbf{B}^1$ have a $q$-target with $(\varepsilon, \delta)$ than from Definition B.1 they can be represented as mixtures. Now observe the if $\mathbf{B}^0 \approx_\varepsilon \mathbf{B}^1$ then the mixtures also satisfy $p \cdot \mathbf{C} + (1 - p) \cdot \mathbf{B}^0 \approx_\varepsilon p \cdot \mathbf{C} + (1 - p) \cdot \mathbf{B}^0$. Using Lemma A.1, we get $\mathbf{Z}^0 \approx_{\varepsilon, \delta} \mathbf{Z}^1$.

For (ii) consider $\mathcal{A}$ and two neighboring datasets $D^0$ and $D^1$. Using Definition B.2 and applying the argument above we obtain $\mathcal{A}(D^0) \approx_{\varepsilon, \delta} \mathcal{A}(D^1)$. The claim follows using Corollary A.3.

Now for the converse. If $\mathbf{Z}^0 \approx_{\varepsilon, \delta} \mathbf{Z}^1$ then consider the decomposition as in Lemma A.1. Now we set $p = 0$ and $\mathbf{B}^b \leftarrow \mathbf{N}^b$ to obtain the claim with $q = 1$ and the target being the full support.

For (ii), if $\mathcal{A} \to \mathcal{Y}$ is $(\varepsilon, \delta)$-DP then consider neighboring $\{D^0, D^1\}$. We have $\mathcal{A}(D^0) \approx_{\varepsilon, \delta} \mathcal{A}(D^1)$. We proceed as with the distributions. $\square$

Algorithm 6 is an extension of Algorithm 1 that permits calls to approximate DP algorithms. The extension also inputs a bound $\tau$ on the number of target hits and a bound $\tau_\delta$ on the cummulative $\delta$ parameter values of the algorithms that were called. We apply adaptively a sequence of $(\varepsilon, \delta)$-DP algorithms with specified $q$-targets to the input data set $D$ and publish the results. We halt when the first of the following happens (1) the respective target sets are hit for a specified $\tau$ number of times (2) the accumulated $\delta$-values exceed the specified limit $\tau_\delta$.

The privacy cost of Target-Charging is as follows (This is a precise and more general statement of Lemma 2.2):

**Algorithm 6:** Target Charging with Approximate DP

**Input:** Dataset $D = \{x_1, \ldots, x_n\} \in X^n$. Integer $\tau \geq 1$ (Upper limit on the number of target hits).
$\quad \tau_\delta \geq 0$ (upper limit on cumulative $\delta$ parameter). Fraction $q \in [0, 1]$.

$C \leftarrow 0, C_\delta \leftarrow 0$                          `// Initialize target hit and failure counters`
**for** $i = 1, \ldots$ **do**                                            `// Main loop`
    Receive $(\mathcal{A}_i, \top_i)$ where $\mathcal{A}_i$ is an $(\varepsilon, \delta_i)$-DP mechanism, and $\top_i$ is a $q$-target with $(\varepsilon, \delta_i)$ for $\mathcal{A}$
    $r \leftarrow \mathcal{A}_i(D)$
    **if** $C_\delta + \delta_i > \tau_\delta$ **then Halt**
    $C_\delta \leftarrow C_\delta + \delta$                               `// TCT charge for `$\delta_i$
    **Publish** $r$
    **if** $r \in \top$ **then**                   `// TCT Charge for a `$q$`-target hit with `$\varepsilon$
        $C \leftarrow C + 1$
        **if** $C = \tau$ **then Halt**

---

**Algorithm 7:** Simulation of Target Charging

**Input:** Two neighboring datasets $D^0, D^1$, private $b \in \{0, 1\}$, $\tau \in \mathbb{N}$, $\tau_\delta \in \mathbb{R}_{\geq 0}$, $q \in [0, 1]$, $\alpha > 0$.

$C \leftarrow 0, C_\delta \leftarrow 0, h \leftarrow 0$    `// Initialize; `$h$` is a counter on the number of non-fail calls to data holder`
**for** $i = 1, \ldots$ **do**                                            `// Main loop`
    Receive $(\mathcal{A}_i, \top_i)$ where $\mathcal{A}_i$ is an $(\varepsilon, \delta_i)$-DP mechanism, and $\top_i$ is a $q$-target with $(\varepsilon, \delta_i)$ for $\mathcal{A}$
    **if** $C_\delta + \delta_i > \tau_\delta$ **then Halt**
    $C_\delta \leftarrow C_\delta + \delta$
    Let $p \in [0, 1]$, $\mathbf{C}, \mathbf{B}^0 \approx_\varepsilon \mathbf{B}^1$, and $\mathbf{E}^b$ (for $b \in \{0, 1\}$) such that
        $\mathcal{A}(D^b) \equiv (1 - \delta) \cdot (p \cdot \mathbf{C} + (1 - p) \cdot \mathbf{B}^b) + \delta \cdot \mathbf{E}^b$          `// By Definition B.2`
    **if** $\mathbf{Ber}(\delta) \equiv 1$ **then**               `// Non-private Data Holder call with Failure`
        **Fail**
        **Publish** $r \sim \mathbf{E}^b$
    **else**
        **if** $\mathbf{Ber}(p) \equiv 1$ **then**
            **Publish** $r \sim \mathbf{C}$                   `// No access to data holder`
        **else**
            **Publish** $r \sim \mathbf{B}^b$                `// `$\varepsilon$`-DP Data Holder Call`
            $h \leftarrow h + 1$               `// counter of `$\varepsilon$`-private data holder calls`
            **if** $h > (1 + \alpha)\tau/q$ **then**       `// Number of Holder calls exceeded limit`
                **Fail**
            **if** $r \in \top$ **then**                  `// outcome is a target hit`
                $C \leftarrow C + 1$
                **if** $C = \tau$ **then Halt**

---

**Theorem B.4** (Privacy of Target-Charging). *Algorithm 6 satisfies the following approximate DP privacy bounds:*

$$\left((1 + \alpha)\frac{\tau}{q}\varepsilon, C_\delta + \delta^*(\tau, \alpha)\right), \qquad\qquad\qquad \textit{for any } \alpha > 0;$$

$$\left(\frac{1}{2}(1 + \alpha)\frac{\tau}{q}\varepsilon^2 + \varepsilon\sqrt{(1 + \alpha)\frac{\tau}{q}\log(1/\delta)}, \delta + C_\delta + \delta^*(\tau, \alpha)\right), \quad \textit{for any } \delta > 0,\ \alpha > 0.$$

*where* $\delta^*(\tau, \alpha) \leq e^{-\frac{\alpha^2}{2(1+\alpha)}\tau}$ *and* $C_\delta \leq \tau_\delta$ *is as computed by the algorithm.*

*Proof.* We apply the simulation-based privacy analysis in Lemma A.5 and use privacy analysis with failure events (Lemma A.6).

The simulation is described in Algorithm 7. Fix two neighboring data sets $D^0$ and $D^1$. The simulator initializes the target hit counter $C \leftarrow 0$ and the cumulative $\delta$-values tracker $C_\delta \leftarrow 0$. For $i \geq 1$ it proceeds as follows. It receives $(\mathcal{A}_i, \top_i)$ where $\mathcal{A}_i$ is $(\varepsilon, \delta_i)$-DP. If $C_\delta + \delta_i > \tau_\delta$ it halts. Since $\top_i$ is a $q$-target for $\mathcal{A}_i$, there are $p, \mathbf{C}, \mathbf{B}^0, \mathbf{B}^1, \mathbf{E}^0$ and $\mathbf{E}^1$ as in Definition B.2. The simulator flips a biased coin $c' \sim \mathbf{Ber}(\delta)$. If $c' = 1$ it outputs $r \sim \mathbf{E}^b$ and the execution is designated as **Fail**. In this

case there is an interaction with the data holder but also a failure designation. The simulator flips a biased coin $c \sim \mathbf{Ber}(p)$. If $c = 1$, then the simulator publishes a sample $r \sim \mathbf{C}$ (this does not require an interaction with the data holder). Otherwise, the data holder is called. The data holder publishes $r \sim \mathbf{B}^b$. We track the number $h$ of calls to the data holder. If $h$ exceeds $(1 + \alpha)\tau/q$, we designate the execution as **Fail**. If $r \in \top_i$ then $C$ is incremented. If $C = \tau$, the algorithm halts.

The correctness of the simulation (faithfully simulating Algorithm 1 on the dataset $D^b$) is straightforward. We analyse the privacy cost. We will show that

  (i) the simulation designated a failure with probability at most $C_\delta + \delta^*(\tau, \alpha)$.

  (ii) Conditioned on no failure designation, the simulation performed at most $r = (1 + \alpha)\frac{\tau}{q}$ adaptive calls to $(\varepsilon, 0)$-DP algorithms

Observe that (ii) is immediate from the simulation declaring failure when $h > r$. We will establish (i) below.

The statement of the Theorem follows from Lemma A.6 and when applying the DP composition bounds (Lemma A.4). The first bounds follow using basic composition and the second follow using advanced composition [14].

This analysis yields the claimed privacy bounds with respect to the private bit $b$. From Lemma A.5 this is the privacy cost of the algorithm.

It remains to show bound the failure probability. There are two ways in which a failure can occur. The first is on each call, with probability $\delta_i$. This probability is bounded by $1 - \prod_i \delta_i \leq \sum_i \delta_i \leq C_\delta$. The second is when the number $h$ of private accesses to the data holder exceeds the limit. We show that the probability that the algorithm halts with failure due to that is at most $\delta^*$.

We consider a process that continues until $\tau$ charges are made. The privacy cost of the simulation (with respect to the private bit $b$) depends on the number of times that the data holder is called. Let $X$ be the random variable that is the number of calls to the data holder. Each call is $\varepsilon$-DP with respect to the private $b$. In each call, there is probability at least $q$ for a "charge" (increment of $C$).

A failure is the event that the number of calls to data holder exceeds $(1 + \alpha)\tau/q$ before $\tau$ charges are made. We show that this occurs with probability at most $\delta^*(\tau, \alpha)$:

$$\Pr\left[X > (1 + \alpha)\frac{\tau}{q}\right] \leq \delta^*(\tau, \alpha) . \tag{1}$$

To establish (1), we first observe that the distribution of the random variable $X$ is dominated by a random variable $X'$ that corresponds to a process of drawing i.i.d. $\mathbf{Ber}(q)$ until we get $\tau$ successes (Domination means that for all $m$, $\Pr[X' > m] \geq \Pr[X > m]$). Therefore, it suffices to establish that

$$\Pr\left[X' > (1 + \alpha)\frac{\tau}{q}\right] \leq \delta^*(\tau, \alpha) .$$

Let $Y$ be the random variable that is a sum of $m = 1 + \left\lfloor (1 + \alpha)\frac{\tau}{q} \right\rfloor$ i.i.d. $\mathbf{Ber}(q)$ random variables. Note that

$$\Pr\left[X' > (1 + \alpha)\frac{\tau}{q}\right] = \Pr[Y < \tau] .$$

We bound $\Pr[Y < \tau]$ using multiplicative Chernoff bounds [4][1]. The expectation is $\mu = mq$ and we bound the probability that the sum of Bernoulli random variables is below $\frac{1}{1+\alpha}\mu = (1 - \frac{\alpha}{1+\alpha})\mu$. Using the simpler form of the bounds we get using $\mu = mq \geq (1 + \alpha)\tau$

$$\Pr[Y < \tau] = \Pr[Y < (1 - \frac{\alpha}{1 + \alpha})\mu] \leq e^{-\frac{\alpha^2}{2(1+\alpha)^2}\mu} \leq e^{-\frac{\alpha^2}{2(1+\alpha)}\tau} .$$

$\square$

---

[1]Bound can be tightened when using precise tail probability values.

**Remark B.5** (Number of target hits)**.** *The TCT privacy analysis has a tradeoff between the final "$\varepsilon$" and "$\delta$" privacy parameters. There is multiplicative factor of $(1 + \alpha)$ ($\sqrt{1 + \alpha}$ with advanced composition) on the "$\varepsilon$" privacy parameter. But when we use a smaller $\alpha$ we need a larger value of $\tau$ to keep the "$\delta$" privacy parameter small. For a given $\alpha, \delta^* > 0$, we can calculate a bound on the smallest value of $\tau$ that works. We get*

$$\tau \geq 2\frac{1 + \alpha}{\alpha^2} \cdot \ln(1/\delta^*) \qquad\qquad \text{(simplified Chernoff)}$$

$$\tau \geq \frac{1}{(1 + \alpha)\ln\left(e^{\alpha/(1+\alpha)}(1 + \alpha)^{-1/(1+\alpha)}\right)} \cdot \ln(1/\delta^*) \qquad\qquad \text{(raw Chernoff)}$$

*For $\alpha = 0.5$ we get $\tau > 10.6 \cdot \ln(1/\delta^*)$. For $\alpha = 1$ we get $\tau > 3.26 \cdot \ln(1/\delta^*)$. For $\alpha = 5$ we get $\tau > 0.31 \cdot \ln(1/\delta^*)$.*

**Remark B.6** (Mix-and-match TCT)**.** *TCT analysis can be extended to the case where we use algorithms with varied privacy guarantees $\varepsilon_i$ and varied $q_i$ values.[2] In this case the privacy cost depends on $\sum_{i|\mathcal{A}_i(D) \in \top_i} \frac{\varepsilon_i}{q_i}$. The analysis relies on tail bounds on the sum of random variables, is more complex. Varied $\varepsilon$ values means the random variables have different size supports. A simple coarse bound is according to the largest support, which allows us to use a simple counter for target hits, but may be lossy with respect to precise bounds. The discussion concerns the (analytical or numerical) derivation of tail bounds is non-specific to TCT and is tangential to our contribution.*

## B.1 Multi-Target TCT

Multi-target charging is described in Algorithm 8. We show the following

**Lemma B.7** (Privacy of multi-TCT)**.** *Algorithm 8 satisfies $(\varepsilon', k\delta')$-approximate DP bounds, where $(\varepsilon', \delta')$ are privacy bounds for single-target charging (Algorithm 1).*

Specifically, when we expect that one (index) of multiple outcomes $\perp_1, \ldots, \perp_k$ will dominate our interaction but can not specify which one it is in advance, we can use $k$-TCT with NotPrior targets with priors $\perp_1, \ldots, \perp_k$. From Lemma B.7, the overall privacy cost depends on the number of times that the reported output is different than the most dominant outcome. More specifically, for private testing, when we expect that one type of outcome would dominate the sequence but we do not know if it is 0 or 1, we can apply 2-TCT. The total number of target hits corresponds to the less dominant outcome. The total number of privacy charges (on average) is at most (approximately for small $\varepsilon$) double that, and therefore is always comparable or better to composition (can be vastly lower when there is a dominant outcome).

---

**Algorithm 8:** Multi-Target Charging

---

**Input:** Dataset $D = \{x_1, \ldots, x_n\} \in X^n$. Integer $\tau \geq 1$ (charging limit). Fraction $q \in [0, 1]$, $k \geq 1$ (number of targets).

**for** $i \in [k]$ **do** $C_i \leftarrow 0$             `// Initialize charge counters`
**while** $\min_{i \in [k]} C_i < \tau$ **do**             `// Main loop`
     **Receive** $(\mathcal{A}, (\top_i)_{i \in [k]})$ where $\mathcal{A}$ is an $\varepsilon$-DP mechanism, and $\top_i$ is a $q$-target for $\mathcal{A}$
     $r \leftarrow \mathcal{A}(D)$
     **Publish** $r$
     **for** $i \in [k]$ **do**
         **if** $r \in \top_i$ **then** $C_i \leftarrow C_i + 1$          `// outcome is in q-target` $\top_i$

---

*Proof of Lemma B.7 (Privacy of multi-Target TCT).* [3] Let $(\varepsilon, \delta)$ be the privacy bounds for $\mathcal{M}_i$ that is single-target TCT with $(\mathcal{A}_i, \top_i)$. Let $\mathcal{M}$ be the $k$-target algorithm. Let $\top_i^j$ be the $i$th target in step $j$.

---

[2]One of our applications of revise calls to conditional release (see Section D applies TCT with both $\varepsilon$-DP and $2\varepsilon$-DP algorithms even for base $\varepsilon$-DP algorithm)

[3]We note that the claim generally holds for online privacy analysis with the best of multiple methods. We provide a proof specific to multi-target charging below.

We say that an outcome sequence $R = (r_j)_{j=1}^h \in R$ is valid for $i \in [k]$ if and only if $\mathcal{M}_i$ would halt with this output sequence, that is, $\sum_{j=1}^h \mathbf{1}\{r_j \in \top_i^j\} = \tau$ and $r_h \in \top_i^h$. We define $G(R) \subset [k]$ to be all $i \in [k]$ for which $R$ is valid.

Consider a set of sequences $H$. Partition $H$ into $k+1$ sets $H_i$ so that $H_0 = \{R \in H \mid G(R) = \emptyset\}$ and $H_i$ may only include $R \in H$ for which $i \in G(R)$. That is, $H_0$ contains all sequences that are not valid for any $i$ and $H_i$ may contain only sequences that are valid for $i$.

$$
\begin{aligned}
\Pr[\mathcal{M}(D) \in H] &= \sum_{i=1}^k \Pr[\mathcal{M}(D) \in H_i] = \sum_{i=1}^k \Pr[\mathcal{M}_i(D) \in H_i] \\
&\leq \sum_{i=1}^k \left(e^\varepsilon \cdot \Pr[\mathcal{M}_i(D') \in H_i] + \delta\right) = e^\varepsilon \cdot \sum_{i=1}^k \Pr[\mathcal{M}_i(D') \in H_i] + k \cdot \delta \\
&= e^\varepsilon \Pr[\mathcal{M}(D') \in H] + k \cdot \delta.
\end{aligned}
$$

$\square$

## C  Properties of `NotPrior` targets

Recall that a `NotPrior` target of an $(\varepsilon, \delta)$-DP algorithm is specified by any potential outcome (of our choice) that we denote by $\bot$. The `NotPrior` target is the set of all outcomes except $\bot$. In this Section we prove (a more general statement of) Lemma 2.3:

**Lemma C.1** (Property of a `NotPrior` target). *Let $\mathcal{M} : X \to \mathcal{Y} \cup \{\bot\}$, where $\bot \notin \mathcal{Y}$, be an $(\varepsilon, \delta)$-DP algorithm. Then the set of outcomes $\mathcal{Y}$ constitutes an $\frac{1}{e^\varepsilon + 1}$-target with $(\varepsilon, \delta)$ for $\mathcal{M}$.*

We will use the following lemma:

**Lemma C.2.** *If two distributions $\mathbf{Z}^0, \mathbf{Z}^1$ with support $\mathcal{Y} \cup \{\bot\}$ satisfy $\mathbf{Z}^0 \approx_\varepsilon \mathbf{Z}^1$ then $\mathcal{Y}$ constitutes an $\frac{1}{e^\varepsilon + 1}$-target with $(\varepsilon, 0)$ for $(\mathbf{Z}^0, \mathbf{Z}^1)$.*

*Proof of Lemma 2.3.* From Definition B.2, it suffices to show that for any two neighboring datasets, $D^0$ and $D^1$, the set $\mathcal{Y}$ is an $\frac{1}{e^\varepsilon + 1}$-target with $(\varepsilon, \delta)$ for $(\mathcal{M}(D^0), \mathcal{M}(D^1))$ (as in Definition B.1).

Consider two neighboring datasets. We have $\mathcal{M}(D^0) \approx_{\varepsilon, \delta} \mathcal{M}(D^1)$. Using Lemma A.1, for $b \in \{0, 1\}$ we can have

$$
\mathcal{M}(D^b) = (1 - \delta) \cdot \mathbf{N}^b + \delta \cdot \mathbf{E}^b, \tag{2}
$$

where $\mathbf{N}^0 \approx_\varepsilon \mathbf{N}^1$. From Lemma C.2, $\mathcal{Y}$ is a $\frac{1}{e^\varepsilon + 1}$-target with $(\varepsilon, 0)$ for $(\mathbf{N}^0, \mathbf{N}^1)$. From Definition B.1 and (2), this means that $\mathcal{Y}$ is a $\frac{1}{e^\varepsilon + 1}$-target with $(\varepsilon, \delta)$ for $(\mathcal{M}(D^0), \mathcal{M}(D^1))$. $\square$

### C.1  Proof of Lemma C.2

We first prove Lemma C.2 for the special case of private testing (when the support is $\{0, 1\}$):

**Lemma C.3** (target for private testing). *Let $\mathbf{Z}^0$ and $\mathbf{Z}^1$ with support $\{0, 1\}$ satisfy $\mathbf{Z}^0 \approx_\varepsilon \mathbf{Z}^1$ Then $\top = \{1\}$ (or $\top = \{0\}$) is an $\frac{1}{e^\varepsilon + 1}$-target with $(\varepsilon, 0)$ for $(\mathbf{Z}^0, \mathbf{Z}^1)$.*

*Proof.* We show that Definition B.1 is satisfied with $\top = \{1\}$, $q = \frac{1}{e^\varepsilon + 1}$ and $(\varepsilon, 0)$, and $\mathbf{Z}^0, \mathbf{Z}^1$.

$$
\begin{aligned}
\pi &= \Pr[\mathbf{Z}^0 \in \top] \\
\pi' &= \Pr[\mathbf{Z}^1 \in \top]
\end{aligned}
$$

be the probabilities of $\top$ outcome in $\mathbf{Z}^0$ and $\mathbf{Z}^1$ respectively. Assume without loss of generality (otherwise we switch the roles of $\mathbf{Z}^0$ and $\mathbf{Z}^1$) that $\pi' \geq \pi$. If $\pi \geq \frac{1}{e^\varepsilon + 1}$, the choice of $p = 0$ and $\mathbf{B}^b = \mathbf{Z}^b$ (and any $\mathbf{C}$) trivially satisfies the conditions of Definition 2.1. Generally, (also for all $\pi < \frac{1}{e^\varepsilon + 1}$):

- Let
$$p = 1 - \frac{\pi' e^\varepsilon - \pi}{e^\varepsilon - 1} \ .$$
Note that since $\mathbf{Z}^0 \approx_\varepsilon \mathbf{Z}^1$ it follows that $\pi' \approx_\varepsilon \pi$ and $(1 - \pi') \approx_\varepsilon (1 - \pi)$ and therefore $p \in [0, 1]$ for any applicable $0 \le \pi \le \pi' \le 1$.

- Let $\mathbf{C}$ be the distribution with point mass on $\perp = \{0\}$.

- Let $\mathbf{B}^0 = \mathbf{Ber}(1 - \frac{\pi' - \pi}{\pi' - e^{-\varepsilon}\pi}) = \mathbf{Ber}(\frac{\pi - \pi e^{-\varepsilon}}{\pi' - e^{-\varepsilon}\pi})$

- Let $\mathbf{B}^1 = \mathbf{Ber}(1 - \frac{\pi' - \pi}{e^\varepsilon \pi' - \pi}) = \mathbf{Ber}(\frac{e^\varepsilon \pi' - \pi'}{e^\varepsilon \pi' - \pi})$

We show that this choice satisfies Definition 2.1 with $q = \frac{1}{e^\varepsilon + 1}$.

- We show that for both $b \in \{0, 1\}$. $\mathbf{Z}^b \equiv p \cdot \mathbf{C} + (1 - p) \cdot \mathbf{B}^b$: It suffices to show that the probability of $\perp$ is the same for the distributions on both sides. For $b = 0$, the probability of $\perp$ in the right hand side distribution is
$$p + (1 - p) \cdot \frac{\pi' - \pi}{\pi' - e^{-\varepsilon}\pi} = 1 - \frac{\pi' e^\varepsilon - \pi}{e^\varepsilon - 1} + \frac{\pi' e^\varepsilon - \pi}{e^\varepsilon - 1} \cdot \frac{\pi' - \pi}{\pi' - e^{-\varepsilon}\pi} = 1 - \pi \ .$$
For $b = 1$, the probability is
$$\begin{aligned} p + (1 - p) \cdot \frac{\pi' - \pi}{e^\varepsilon \pi' - \pi} &= 1 - \frac{\pi' e^\varepsilon - \pi}{e^\varepsilon - 1} + \frac{\pi' e^\varepsilon - \pi}{e^\varepsilon - 1} \cdot \frac{\pi' - \pi}{e^\varepsilon \pi' - \pi} \\ &= 1 - \frac{\pi' e^\varepsilon - \pi}{e^\varepsilon - 1} \left(1 - \frac{\pi' - \pi}{e^\varepsilon \pi' - \pi}\right) \\ &= 1 - \frac{\pi' e^\varepsilon - \pi}{e^\varepsilon - 1} \cdot \frac{e^\varepsilon \pi' - \pi - \pi' + \pi}{e^\varepsilon \pi' - \pi} = 1 - \pi' \ . \end{aligned}$$

- We show that for $b \in \{0, 1\}$, $\Pr[\mathbf{B}^b \in \top] \ge \frac{1}{e^\varepsilon + 1}$.
$$\begin{aligned} \Pr[\mathbf{B}^0 \in \top] &= \frac{\pi - e^{-\varepsilon}\pi}{\pi' - e^{-\varepsilon}\pi} \\ &\ge \frac{\pi - e^{-\varepsilon}\pi}{e^\varepsilon \pi - e^{-\varepsilon}\pi} = \frac{e^\varepsilon - 1}{e^{2\varepsilon} - 1} = \frac{1}{e^\varepsilon + 1} \ . \\ \Pr[\mathbf{B}^1 \in \top] &= \frac{\pi'(e^\varepsilon - 1)}{\pi' e^\varepsilon - \pi} \\ &\ge \frac{\pi(e^\varepsilon - 1)}{\pi e^{2\varepsilon} - \pi} = \frac{e^\varepsilon - 1}{e^{2\varepsilon} - 1} = \frac{1}{e^\varepsilon + 1} \end{aligned}$$
Note that the inequalities are tight when $\pi' = \pi$ (and are tighter when $\pi'$ is closer to $\pi$). This means that our selected $q$ is the largest possible that satisfies the conditions for the target being $\top$.

- We show that $\mathbf{B}^0$ and $\mathbf{B}^1$ are $\varepsilon$-indistinguishable, that is
$$\mathbf{Ber}(1 - \frac{\pi' - \pi}{\pi' - e^{-\varepsilon}\pi}) \approx_\varepsilon \mathbf{Ber}(1 - \frac{\pi' - \pi}{e^\varepsilon \pi' - \pi}).$$
Recall that $\mathbf{Ber}(a) \approx_\varepsilon \mathbf{Ber}(b)$ if and only if $a \approx_\varepsilon b$ and $(1 - a) \approx_\varepsilon (1 - b)$. First note that
$$e^{-\varepsilon} \cdot \frac{\pi' - \pi}{\pi' - e^{-\varepsilon}\pi} = \frac{\pi' - \pi}{e^\varepsilon \pi' - \pi}$$
Hence
$$\frac{\pi' - \pi}{\pi' - e^{-\varepsilon}\pi} \approx_\varepsilon \frac{\pi' - \pi}{e^\varepsilon \pi' - \pi} \ .$$
It also holds that
$$1 \le \frac{\frac{\pi - e^{-\varepsilon}\pi}{\pi' - e^{-\varepsilon}\pi}}{\frac{\pi'(1 - e^{-\varepsilon})}{\pi' - e^{-\varepsilon}\pi}} = \frac{\pi'}{\pi} \le e^\varepsilon.$$

$\square$

*Proof of Lemma C.2.* The proof is very similar to that of Lemma C.3, with a few additional details since $\top = \mathcal{Y}$ can have more than one element (recall that $\bot$ is a single element).

Assume (otherwise we switch roles) that $\Pr[\mathbf{Z^0} = \bot] \geq \Pr[\mathbf{Z^1} = \bot]$. Let

$$\pi = \Pr[\mathbf{Z^0} \in \mathcal{Y}]$$
$$\pi' = \Pr[\mathbf{Z^1} \in \mathcal{Y}]\,.$$

Note that $\pi' \geq \pi$.

We choose $p$, $\mathbf{C}$, $\mathbf{B}^0$, $\mathbf{B}^1$ as follows. Note that when $\pi \geq \frac{1}{e^\varepsilon+1}$, then the choice of $p = 0$ and $\mathbf{B}^b = \mathbf{Z}^b$ satisfies the conditions. Generally,

- Let

$$p = 1 - \frac{\pi'e^\varepsilon - \pi}{e^\varepsilon - 1}\,.$$

- Let $\mathbf{C}$ be the distribution with point mass on $\bot$.

- Let $\mathbf{B}^0$ be $\bot$ with probability $\frac{\pi'-\pi}{\pi'-e^{-\varepsilon}\pi}$ and otherwise (with probability $\frac{\pi-\pi e^{-\varepsilon}}{\pi'-e^{-\varepsilon}\pi}$) be $\mathbf{Z}^0$ conditioned on the outcome being in $\mathcal{Y}$.

- Let $\mathbf{B}^1$ be $\bot$ with probability $\frac{\pi'-\pi}{e^\varepsilon \pi'-\pi}$ and otherwise (with probability $\frac{e^\varepsilon \pi'-\pi'}{e^\varepsilon \pi'-\pi}$) be $\mathbf{Z}^1$ conditioned on the outcome being in $\mathcal{Y}$.

It remains to show that these choices satisfy Definition 2.1:

The argument for $\Pr[\mathbf{B}^b \in \mathcal{Y}] \geq \frac{e^\varepsilon-1}{e^{2\varepsilon}-1}$ is identical to Lemma C.3 (with $\mathcal{Y} = \top$).

We next verify that for $b \in \{0,1\}$: $\mathbf{Z}^b \equiv p \cdot \mathbf{C} + (1-p) \cdot \mathbf{B}^b$. The argument for the probability of $\bot$ is identical to Lemma C.3. The argument for $y \in \mathcal{Y}$ follows from the probability of being in $\mathcal{Y}$ being the same and that proportions are maintained.

For $b = 0$, the probability of $y \in \mathcal{Y}$ in the right hand side distribution is

$$(1-p) \cdot \frac{\pi - \pi e^{-\varepsilon}}{\pi' - e^{-\varepsilon}\pi} \cdot \frac{\Pr[\mathbf{Z}^0 = y]}{\Pr[\mathbf{Z}^0 \in \mathcal{Y}]} = \pi \cdot \frac{\Pr[\mathbf{Z}^0 = y]}{\Pr[\mathbf{Z}^0 \in \mathcal{Y}]} = \Pr[\mathbf{Z}^0 = y].$$

For $b = 1$, the probability of $y \in \mathcal{Y}$ in the right hand side distribution is

$$(1-p) \cdot \frac{e^\varepsilon \pi' - \pi'}{e^\varepsilon \pi' - \pi} \cdot \frac{\Pr[\mathbf{Z}^1 = y]}{\Pr[\mathbf{Z}^1 \in \mathcal{Y}]} = \pi' \cdot \frac{\Pr[\mathbf{Z}^1 = y]}{\Pr[\mathbf{Z}^1 \in \mathcal{Y}]}$$
$$= \Pr[\mathbf{Z}^1 = y].$$

Finally, we verify that $\mathbf{B}^0$ and $\mathbf{B}^1$ are $\varepsilon$-indistinguishable. Let $W \subset \mathcal{Y}$. We have

$$\Pr[\mathbf{B}^0 \in W] = \frac{\pi(1 - e^{-\varepsilon})}{\pi' - e^{-\varepsilon}\pi} \cdot \frac{\Pr[\mathbf{Z}^0 \in W]}{\pi} = \frac{e^\varepsilon - 1}{\pi'e^\varepsilon - \pi}\Pr[\mathbf{Z}^0 \in W]$$
$$\Pr[\mathbf{B}^1 \in W] = \frac{\pi'(e^\varepsilon - 1)}{e^\varepsilon \pi' - \pi} \cdot \frac{\Pr[\mathbf{Z}^1 \in W]}{\pi'} = \frac{e^\varepsilon - 1}{\pi'e^\varepsilon - \pi}\Pr[\mathbf{Z}^1 \in W]\,.$$

Therefore

$$\frac{\Pr[\mathbf{B}^0 \in W]}{\Pr[\mathbf{B}^1 \in W]} = \frac{\Pr[\mathbf{Z}^0 \in W]}{\Pr[\mathbf{Z}^1 \in W]}$$

and we use $\mathbf{Z}^0 \approx_\varepsilon \mathbf{Z}^1$. The case of $W = \bot$ is identical to the proof of Lemma C.3. The case $\bot \in W$ follows. $\square$

# D  Conditional Release with Revisions

In this section we analyze an extension to conditional release that allows for revision calls to be made with respect to *previous* computations. This extension was presented in Section 2.3 and described in Algorithm 2. A conditional release applies a private algorithm $\mathcal{A} \to \mathcal{Y}$ with respect to a subset of outcomes $\top \subset \mathcal{Y}$. It draws $y \sim \mathcal{A}(D)$ and returns $y$ if $y \in \top$ and $\bot$ otherwise. Each revise calls effectively expands the target to $\top_h \cup \top'$, when $\top_h$ is the prior target and $\top'$ a disjoint extension. If the (previously) computed result hits the expanded target ($y \in \top'$), the value $y$ is reported and charged. Otherwise, additional revise calls can be performed. The revise calls can be interleaved with other TCT computations at any point in the interaction.

## D.1  Preliminaries

For a distribution $\mathbf{Z}$ with support $\mathcal{Y}$ and $W \subset \mathcal{Y}$ we denote by $\mathbf{Z}_W$ the distribution with support $W \cup \{\bot\}$ where outcomes not in $W$ are "replaced" by $\bot$. That is, for $y \in W$, $\Pr[\mathbf{Z}_W = y] := \Pr[\mathbf{Z} = y]$ and $\Pr[\mathbf{Z}_W = \bot] := \Pr[\mathbf{Z} \notin W]$.

For a distribution $\mathbf{Z}$ with support $\mathcal{Y}$ and $W \subset \mathcal{Y}$ we denote by $\mathbf{Z} \mid W$ the *conditional distribution* of $\mathbf{Z}$ on $W$. That is, for $y \in W$, $\Pr[(\mathbf{Z} \mid W) = y] := \Pr[\mathbf{Z} = y] / \Pr[\mathbf{Z} \in W]$.

**Lemma D.1.** *If $\mathbf{B}^0 \approx_{\varepsilon,\delta} \mathbf{B}^1$ then $\mathbf{B}^0_W \approx_{\varepsilon,\delta} \mathbf{B}^1_W$.*

**Lemma D.2.** *Let $\mathbf{B}^0$, $\mathbf{B}^1$ be probability distributions with support $\mathcal{Y}$ such that $\mathbf{B}^0 \approx_\varepsilon \mathbf{B}^1$. Let $W \subset \mathcal{Y}$. Then $\mathbf{B}^0 \mid W \approx_{2\varepsilon} \mathbf{B}^1 \mid W$.*

We extend these definitions to a randomized algorithm $\mathcal{A}$, where $\mathcal{A}_W(D)$ has distribution $\mathcal{A}(D)_W$ and $(\mathcal{A} \mid W)(D)$ has distribution $\mathcal{A}(D) \mid W$. The claims in Lemma D.1 and Lemma D.2 then transfer to privacy of the algorithms.

## D.2  Analysis

To establish correctness, it remains to show that each `ConditionalRelease` call with an $(\varepsilon, \delta)$-DP algorithm $\mathcal{A}$ can be casted in TCT as a call to an $(\varepsilon, \delta)$-DP algorithm with a `NotPrior` target and each `ReviseCR` call cap be casted as a call to an $2\varepsilon$-DP algorithm with a `NotPrior` target.

*Proof of Lemma 2.5.* The claim for `ConditionalRelease` was established in Lemma 2.4: Conditional release `ConditionalRelease` $(\mathcal{A}, \top)$ calls the algorithm $\mathcal{A}_\top$ with target $\top$. From Lemma D.1, $\mathcal{A}_\top$ is $(\varepsilon, \delta)$-DP when $\mathcal{A}$ is $(\varepsilon, \delta)$-DP. $\top$ constitutes a `NotPrior` target for $\mathcal{A}_\top$ with respect to prior $\bot$.

We next consider revision calls as described in Algorithm 2. We first consider the case of a pure-DP $\mathcal{A}$ ($\delta = 0$).

When `ConditionalRelease` publishes $\bot$, the internally stored value $r_h$ conditioned on published $\bot$ is a sample from the conditional distribution $\mathcal{A}(D) \mid \neg\top$.

We will show by induction that this remains true after `ReviseCR` calls, that is the distribution of $r_h$ conditioned on $\bot$ being returned in all previous calls is $\mathcal{A}(D) \mid \neg\top_h$ where $\top_h$ is the current expanded target.

An `ReviseCR` call with respect to current target $\top_h$ and extension $\top'$ can be equivalently framed as drawing $r \sim \mathcal{A}(D) \mid \neg\top_h$. From Lemma D.2, if $\mathcal{A}$ is $\varepsilon$-DP then $\mathcal{A} \mid \neg\top_h$ is $2\varepsilon$-DP. If $r \in \top'$ we publish it and otherwise we publish $\bot$. This is a conditional release computation with respect to the $2\varepsilon$-DP algorithm $\mathcal{A} \mid \neg\top_h$ and the target $\top'$. Equivalently, it is a call to the $2\varepsilon$-DP algorithm $(\mathcal{A} \mid \neg\top_h)_{\top'}$ with a `NotPrior` target $\top'$.

Following the `ReviseCR` call, the conditional distribution of $r_h$ conditioned on $\bot$ returned in the previous calls is $\mathcal{A}(D) \mid \neg(\top_h \cup \top')$ as claimed. We then update $\top_h \leftarrow \top_h \cup \top'$.

It remains to handle the case $\delta > 0$. We consider `ReviseCR` calls for the case where $\mathcal{A}$ is $(\varepsilon, \delta)$-DP (approximate DP). In this case, we want to show that we charge for the $\delta$ value once, only on the original `ConditionalRelease` call. We apply the simulation-based analysis in the proof of Theorem B.4 with two fixed neighboring datasets. Note that this can be viewed as each call being

with a pair of distributions with an appropriate $q$-target (that in our case is always a `NotPrior` target).

The first `ConditionalRelease` call uses the distributions $\mathcal{A}(D^0)$ and $\mathcal{A}(D^1)$. From Lemma A.1 they can be expressed as respective mixtures of pure $\mathbf{N}^0 \approx_\varepsilon \mathbf{N}^1$ part (with probability $1 - \delta$) and non-private parts. The non-private draw is designated failure with probability $\delta$. Effectively, the call in the simulation is then applied to the pair $(\mathbf{N}^0_\top, \mathbf{N}^1_\top)$ with target $\top$.

A followup `ReviseCR` call is with respect to the previous target $\top_h$ and target extension $\top'$. The call is with the distributions $(\mathbf{N}^b \mid \neg\top_h)_{\top'}$ that using Lemma D.1 and Lemma D.2 satisfy $(\mathbf{N}^0 \mid \neg\top_h)_{\top'} \approx_{2\varepsilon} (\mathbf{N}^1 \mid \neg\top_h)_{\top'}$. $\qquad\square$

# E   Boundary Wrapper Analysis

In this section we provide details for the boundary wrapper method including proofs of Lemma 2.7 and Lemma 2.8. For instructive reasons, we first consider the special case of private testing and then outline the extensions to private classification.

Algorithm 4 when specialized for tests first computes $\pi(D) = \min\{\Pr[\mathcal{A}(D) = 0], 1 - \Pr[\mathcal{A}(D) = 0]\}$, returns $\top$ with probability $\pi/(1 + \pi)$ and otherwise (with probability $1/(1 + \pi)$) return $\mathcal{A}(D)$. Overall, we return the less likely outcome with probability $\pi/(1 + \pi)$, and the more likely one with probability $(1 - \pi)/(1 + \pi)$.

**Lemma E.1** (Privacy of wrapped test). *If the test is $\varepsilon$-DP then the wrapper test is $t(\varepsilon)$-DP where $t(\varepsilon) \leq \frac{4}{3}\varepsilon$.*

*Proof.* Working directly with the definitions, $t(\varepsilon)$ is the maximum of

$$\max_{\pi \in (0, 1/2)} \left| \ln\left( \frac{1 - e^{-\varepsilon}\pi}{1 + e^{-\varepsilon}\pi} \cdot \frac{1 + \pi}{1 - \pi} \right) \right| \leq \frac{4}{3}\varepsilon \tag{3}$$

$$\max_{\pi \in (0, 1/2)} \left| \ln\left( \frac{e^{-\varepsilon}\pi}{1 + e^{-\varepsilon}\pi} \cdot \frac{1 + \pi}{\pi} \right) \right| \leq \varepsilon \tag{4}$$

$$\max_{\pi \in (\frac{e^{-\varepsilon}}{2}, \frac{1}{1+e^\varepsilon})} \left| \ln\left( \frac{\pi}{1 + \pi} \cdot \frac{2 - e^\varepsilon\pi}{e^\varepsilon\pi} \right) \right| \leq \varepsilon \tag{5}$$

$$\max_{\pi \in (\frac{e^{-\varepsilon}}{2}, \frac{1}{1+e^\varepsilon})} \left| \ln\left( \frac{1 - \pi}{1 + \pi} \cdot \frac{2 - e^\varepsilon\pi}{1 - e^\varepsilon\pi} \right) \right| \leq \frac{4}{3}\varepsilon \tag{6}$$

$$\max_{\pi \in (\frac{e^{-\varepsilon}}{2}, \frac{1}{1+e^\varepsilon})} \left| \ln\left( \frac{\pi}{1 + \pi} \cdot \frac{2 - e^\varepsilon\pi}{1 - e^\varepsilon\pi} \right) \right| \leq \varepsilon \tag{7}$$

Inequality (3) bounds the ratio change in the probably of the larger probability outcome when it remains the same and (4) the ratio change in the probability of the smaller probability outcome when it remains the same between the neighboring datasets. When the less probable outcome changes between the neighboring datasets it suffices to consider the case where the probability of the initially less likely outcome changes to $e^\varepsilon\pi > 1/2$ so that $e^\varepsilon\pi < 1 - \pi$, that is the change is from $\pi$ to $e^\varepsilon\pi$ where $\pi \in (\frac{e^{-\varepsilon}}{2}, \frac{1}{1+e^\varepsilon})$. Inequalities 5 and 6 correspond to this case. The wrapped probabilities of the $\top$ outcome are the same as the less probably outcome in the case that it is the same in the two databases. Inequality 7 corresponds to the case when there is change. $\qquad\square$

We now show that $\top$ is a target for the wrapped test.

**Lemma E.2** ($q$-value of the boundary target). *The outcome $\top$ of a boundary wrapper of an $\varepsilon$-DP test is a $\frac{e^{t(\varepsilon)} - 1}{2(e^{\varepsilon + t(\varepsilon)} - 1)}$-target.*

*Proof.* Consider two neighboring datasets where the same outcome is less likely for both and $\pi \leq \pi'$. Suppose without loss of generality that 0 is the less likely outcome.

The common distribution $(C)$ has point mass on 1.

The distribution $\mathbf{B}^0$ is a scaled part of $\mathcal{M}(D^0)$ that includes all 0 and $\top$ outcomes (probability $\pi/(1+\pi)$ each) and probability of $\Delta \frac{e^{t(\varepsilon)}}{e^{t(\varepsilon)}-1}$ of the 1 outcomes, where $\Delta = \frac{2\pi'}{1+\pi'} - \frac{2\pi}{1+\pi}$.

The distribution $\mathbf{B}^1$ is a scaled part of $\mathcal{M}(D^1)$ that includes all 0 and $\top$ outcomes (probability $\pi'/(1+\pi')$ each) and probability of $\Delta \frac{1}{e^{t(\varepsilon)}-1}$ of the 1 outcomes.

It is easy to verify that $\mathbf{B}^0 \approx_{t(\varepsilon)} \mathbf{B}^1$ and that

$$
\begin{aligned}
1 - p &= \frac{2\pi'}{1+\pi'} + \Delta \frac{1}{e^{t(\varepsilon)}-1} = \frac{2\pi}{1+\pi} + \Delta \frac{e^{t(\varepsilon)}}{e^{t(\varepsilon)}-1} \\
&= \frac{2\pi'}{1+\pi'} \frac{e^{t(\varepsilon)}}{e^{t(\varepsilon)}-1} - \frac{2\pi}{1+\pi} \frac{1}{e^{t(\varepsilon)}-1} \\
&= \frac{2}{e^{t(\varepsilon)}-1} \left( e^{t(\varepsilon)} \frac{\pi'}{1+\pi'} - \frac{\pi}{1+\pi} \right)
\end{aligned}
$$

Using $\frac{\pi}{1+\pi} \le \frac{\pi'}{1+\pi'}$ and $\frac{\frac{\pi'}{1+\pi'}}{\frac{\pi}{1+\pi}} \le e^{\varepsilon}$ we obtain

$$
\begin{aligned}
q &\ge \frac{\frac{\pi}{1+\pi}}{1-p} \\
&= \frac{e^{t(\varepsilon)}-1}{2} \left( \frac{1}{e^{t(\varepsilon)} \cdot \frac{\pi'}{1+\pi'} \cdot \frac{1+\pi}{\pi} - 1} \right) \\
&\ge \frac{e^{t(\varepsilon)}-1}{2} \frac{1}{e^{t(\varepsilon)+\varepsilon}-1}.
\end{aligned}
$$

$\square$

**Extension to Private Classification**  To extension from Lemma E.1 to Lemma 2.7 follows by noting that the same arguments also hold respectively for sets of outcomes and also cover the case when there is no dominant outcome and when there is a transition between neighboring datasets from no dominant outcome to a dominant outcome. The extension from Lemma E.1 to Lemma 2.7 is also straightforward by also noting the cases above (that only make the respective $\Delta$ smaller), and allowing $\mathbf{C}$ to be empty when there is no dominant outcome.

## F   Boundary wrapping without a probability oracle

We present a boundary-wrapping method that does not assume a probability oracle. This method accesses the distribution $\mathcal{A}(D)$ in a blackbox fashion.

At a very high level, we show that one can run an $(\varepsilon, 0)$-DP algorithm $\mathcal{A}$ twice and observe both outcomes. Then, denote by $\mathcal{Y}$ the range of the algorithm $\mathcal{A}$. We can show that $E = \{(y, y') : y \ne y'\} \subseteq \mathcal{Y} \times \mathcal{Y}$ is an $\Omega(1)$-target of this procedure. That is, if the analyst observes the same outcome twice, she learns the outcome "for free". If the two outcomes are different, the analyst pays $O(\varepsilon)$ of privacy budget, but she will be able to access both outcomes, which is potentially more informative than a single execution of the algorithm.

**Lemma F.1.** *Suppose $\mathcal{A} : \mathcal{X}^* \to \mathcal{Y}$ is an $(\varepsilon, 0)$-DP algorithm where $|\mathcal{Y}| < \infty$. Denote by $\mathcal{A} \circ \mathcal{A}$ the following algorithm: on input $D$, independently run $\mathcal{A}$ twice and publish both outcomes. Define $E := \{(y, y') : y \ne y'\} \subseteq \mathcal{Y} \times \mathcal{Y}$. Then, $\mathcal{A} \circ \mathcal{A}$ is a $(2\varepsilon, 0)$-DP algorithm, and $E$ is a $f(\varepsilon)$-target for $\mathcal{A} \circ \mathcal{A}$, where*

$$
f(\varepsilon) = 1 - \sqrt{e^{2\varepsilon}/(1+e^{2\varepsilon})}.
$$

*Proof.* $\mathcal{A} \circ \mathcal{A}$ is $(2\varepsilon, 0)$-DP by the basic composition theorem. Next, we verify the second claim.

Identify elements of $\mathcal{Y}$ as $1, 2, \ldots, m = |\mathcal{Y}|$. Let $D, D'$ be two adjacent data sets. For each $i \in [m]$, let

$$
p_i = \Pr[\mathcal{A}(D) = i], \quad p_i' = \Pr[\mathcal{A}(D') = i].
$$

We define a distribution $\mathbf{C}$. For each $i \in [m]$, define $q_i$ to be the largest real such that

$$p_i^2 - q_i \in [e^{-2\varepsilon}(p_i'^2 - q_i), e^{2\varepsilon}(p_i'^2 - q_i)].$$

Then, we define $\mathbf{C}$ to be a distribution over $\{(i,i) : i \in [m]\}$ where $\Pr[\mathbf{C} = (i,i)] = \frac{q_i}{\sum_j q_j}$.

We can then write $(\mathcal{A} \circ \mathcal{A})(D) = \alpha \cdot \mathbf{C} + (1 - \alpha) \cdot \mathbf{N}^0$ and $(\mathcal{A} \circ \mathcal{A})(D') = \alpha \cdot \mathbf{C} + (1 - \alpha) \cdot \mathbf{N}^1$, where $\alpha = \sum_i q_i$, and $\mathbf{N}^0$ and $\mathbf{N}^1$ are $2\varepsilon$-indistinguishable.

Next, we consider lower-bounding $\Pr[\mathbf{N}^0 = (y, y') : y \neq y']$. The lower bound of $\Pr[\mathbf{N}^0 = (y, y') : y \neq y']$ will follow from the same argument.

Indeed, we have

$$\frac{\Pr[\mathbf{N}^0 = (y, y') : y \neq y']}{\Pr[\mathbf{N}^0 = (y, y)]} = \frac{\sum_i p_i(1 - p_i)}{\sum_i p_i^2 - q_i}.$$

We claim that

$$p_i^2 - q_i \leq 1 - p_i'^2.$$

The inequality is trivially true if $p_i^2 \leq 1 - p_i'^2$. Otherwise, we can observe that for $q := p_i^2 + p_i'^2 - 1 > 0$, we have $p_i^2 - q = 1 - p_i'^2$ and $p_i'^2 - q = 1 - p_i^2$. Since $1 - p_i'^2 \in [e^{-2\varepsilon}(1 - p_i^2), e^{2\varepsilon}(1 - p_i^2)]$, this implies that $q_i$ can only be larger than $q$.

Since we also trivially have that $p_i^2 - q_i \leq p_i^2$, we conclude that

$$\frac{\Pr[\mathbf{N}^0 = (y, y') : y \neq y']}{\Pr[\mathbf{N}^0 = (y, y)]} \geq \frac{\sum_i p_i(1 - p_i)}{\sum_i \min(p_i^2, 1 - p_i'^2)} \geq \frac{\sum_i p_i(1 - p_i)}{\sum_i \min(p_i^2, e^{2\varepsilon}(1 - p_i^2))}.$$

Next, it is straightforward to show that, for every $p \in [0, 1]$, one has

$$\frac{p(1 - p)}{\min(p^2, e^{2\varepsilon}(1 - p^2))} = \min\left(\frac{1 - p}{p}, \frac{p}{e^{2\varepsilon}(1 + p)}\right) \geq \frac{1 - \sqrt{e^{2\varepsilon}/(1 + e^{2\varepsilon})}}{\sqrt{e^{2\varepsilon}/(1 + e^{2\varepsilon})}}.$$

Consequently,

$$\Pr[\mathbf{N}^0 = (y, y') : y \neq y'] = \frac{\Pr[\mathbf{N}^0 = (y, y') : y \neq y']}{\Pr[\mathbf{N}^0 = (y, y') : y \neq y'] + \Pr[\mathbf{N}^0 = (y, y)]} \geq 1 - \sqrt{e^{2\varepsilon}/(1 + e^{2\varepsilon})},$$

as desired. $\qquad\square$

**Remark F.2.** *For a typical use case where $\epsilon = 0.1$, we have $f(\varepsilon) \approx 0.258$. Then, by applying Theorem B.4, on average we pay $\approx 8\varepsilon$ privacy cost for each target hit. Improving the constant of 8 is a natural question for future research. We also note that while the overhead is more significant compared to the boundary wrapper of Algorithm 4, the output is more informative as it includes two independent responses of the core algorithm whereas Algorithm 4 returns one or none (when $\top$ is returned). We expect that it is possible to design less-informative boundary wrappers for the case of blackbox access (no probability oracle) that have a lower overhead. We leave this as an interesion question for followup work.*

## G  $q$ value for `BetweenThresholds`

We provide details for the `BetweenThresholds` classifier (see Section 2.6). The `BetweenThresholds` classifier is a refinement of `AboveThreshold`. It is specified by a 1-Lipschitz function $f$, two thresholds $t_\ell < t_r$, and a privacy parameter $\varepsilon$. We compute $\tilde{f}(D) = f(D) + \mathbf{Lap}(1/\varepsilon)$, where $\mathbf{Lap}$ is the Laplace distribution. If $\tilde{f}(D) < t_\ell$ we return L. If $\tilde{f}(D) > t_r$ we return H. Otherwise, we return $\top$.

**Lemma G.1** (Effectiveness of the "between" target)**.** *The $\top$ outcome is an $(1 - e^{-(t_r - t_l)\varepsilon}) \cdot \frac{e^\varepsilon - 1}{e^{2\varepsilon} - 1}$-target for* `BetweenThresholds`*.*

*Proof.* Without loss of generality we assume that $t_\ell = 0$ and $t_r = t/\varepsilon$ for a parameter $t$.

Consider two neighboring data sets $D^0$ and $D^1$ and the respective $f(D^0)$ and $f(D^1)$. Since $f$ is 1-Lipschitz, we can assume without loss of generality (otherwise we switch the roles of the two data

sets) that $f(D^0) \leq f(D^1) \leq f(D^0) + 1$. Consider the case $f(D^1) \leq 0$. The case $f(D^0) \geq t/\varepsilon$ is symmetric and the cases where one or both of $f(D^b)$ are in $(0, t/\varepsilon)$ make $\perp$ a more effective target.

$$\pi_L^b := \Pr[f(D^b) + \mathbf{Lap}(1/\varepsilon) < t_\ell = 0] = 1 - \frac{1}{2}e^{-|f(D^b)|\varepsilon}$$

$$\pi_H^b := \Pr[f(D^b) + \mathbf{Lap}(1/\varepsilon) > t_r = t/\varepsilon] = \frac{1}{2}e^{-(|f(D^b)|\varepsilon - t}$$

$$\pi_\top^b := \Pr[f(D^b) + \mathbf{Lap}(1/\varepsilon) \in (0, t/\varepsilon)] = \frac{1}{2}\left(e^{-|f(D^b)|\varepsilon} - e^{-(|f(D^b)|\varepsilon - t)}\right) = \frac{1}{2}e^{-|f(D^b)|\varepsilon}(1 - e^{-t})$$

Note that $\pi_L^0 \approx_\varepsilon \pi_L^1$ and $\pi_H^1 \approx_\varepsilon \pi_H^0$, $\pi_L^0 \geq \pi_L^1$ and $\pi_H^1 \geq \pi_H^0$

We set

$$p = (\pi_L^1 - \frac{1}{e^\varepsilon - 1}(\pi_L^0 - \pi_L^1)) + (\pi_H^0 - \frac{1}{e^\varepsilon - 1}(\pi_H^1 - \pi_H^0))$$

and the distribution $\mathbf{C}$ to be L with probability $(\pi_L^1 - \frac{1}{e^\varepsilon - 1}(\pi_L^0 - \pi_L^1))/p$ and H otherwise.

We specify $p$ and the distributions $\mathbf{B}^b$ and $\mathbf{C}$ as we did for `NotPrior` (Lemma 2.3) with respect to "prior" L. (We can do that and cover also the case where $f(D^0) > t/\varepsilon$ where the symmetric prior would be H because the target does not depend on the values being below or above the threshold).

The only difference is that our target is smaller, and includes only $\top$ rather than $\top$ and H. Because of that, the calculated $q$ value is reduced by a factor of

$$\frac{\pi_\top^b}{\pi_\top^b + \pi_H^b} = \frac{\frac{1}{2}e^{-|f(D^b)|\varepsilon}(1 - e^{-t})}{\frac{1}{2}e^{-|f(D^b)|\varepsilon}} = (1 - e^{-t}) .$$

$\square$

# H   Numerical Comparison of `BetweenThresholds`

We demonstrate numerically the benefits of our TCT-based `BetweenThresholds` (Section 2.6) compared with the SVT-based prior work of [3].

Consider a sensitive dataset $D = \{x_1, \ldots, x_n\}$ of size $n$ and thresholds $t_l < t_r$. We process a sequence of queries where each is specified by a function $f : X \rightarrow [0, 1]$. We aim to process each query by performing a noisy comparison of $\frac{1}{n}\sum_{i=1}^n f(x_i)$ to $t_l$ and $t_r$ with the goal of having a error of at most $\alpha$ on our responses to queries with confidence at least $\beta = 1 - 1/n$. That is, for `below` responses it holds that $\frac{1}{n}\sum_{i=1}^n f(x_i) \leq t_l + \alpha$, for `above` responses it holds that $\frac{1}{n}\sum_{i=1}^n f(x_i) \geq t_r - \alpha$, and for `between` responses it holds that $t_l - \alpha \leq \frac{1}{n}\sum_{i=1}^n f(x_i) \leq t_r + \alpha$.

We then consider how many $\top$ (that is, `between`) responses we can incur under the requirement that the whole sequence is $(\varepsilon, \delta)$-DP with $\varepsilon \leq 1$ and $\delta = 1/n$.

**TCT:** For a query $f$, we compare the noisy count $\frac{1}{n}\sum_{i=1}^n f(x_i) + \mathbf{Lap}(\frac{\alpha}{\log(n)})$ with the thresholds $t_l, t_r$ and report `below`, `between` ($\top$), or `above` accordingly. To achieve the desired accuracy and confidence of $\alpha$ and $\beta = 1 - 1/n$, we set the Laplace noise parameter to $\frac{\alpha}{\log(n)}$. This ensures that with confidence at least $(1 - \frac{1}{n})$, the noisy count is within relative error $\alpha$ of the true count. The sensitivity is $1/n$ and each call is $\varepsilon'$-DP with $\varepsilon' = \frac{\log n}{\alpha n}$.

For $t_r - t_l \geq \frac{2}{\varepsilon'}$, each call has a $q$-target with $q = \frac{1 - e^{-2}}{1 + e^{\varepsilon'}}$. We can then compute the overall privacy cost for a number of target hits using basic or advanced composition according to Theorem B.4 using the appropriate $q$ values and $\delta$. We solve for the number of target hits that keeps the overall privacy parameter values below $(1, 1/n)$.

**SVT-based [3]:** Through the analysis in [3], we obtain that we need $\varepsilon' \geq \frac{16 \cdot \log(n)}{\alpha n}$. An "optimistic" analysis that we performed implied that we can not get better than $\varepsilon' \geq \frac{4 \cdot \log(n)}{\alpha n}$.

Figure 1 shows the dependence of the number of `between` responses ($y$ axis) on the number of data points $n$ ($x$ axis) for TCT and for the provided and optimistic bounds for [3]. TCT has a factor of 95 gain with respect to the provided bound and a factor of 6 gain over the optimistic bound.

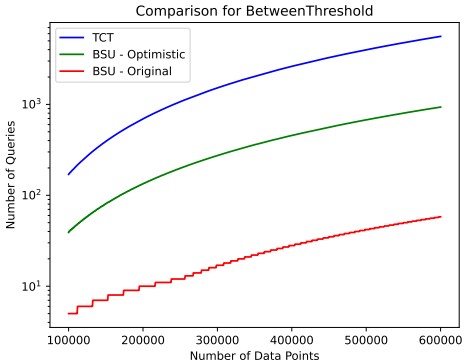

Figure 1: Number of `BetweenThresholds` target hits for data set size. $\delta = \beta = 1/n$, $\alpha = 0.01$

Finally, the gains reported in the figure assume that both algorithms use the same gap that is large enough to satisfy the requirement of [3] that $|t_r - t_l| >= \frac{12(\log(1/\varepsilon') + \log(1/\delta))}{\varepsilon'}$. But in situations where we are interested in a narrow gap around a single threshold, TCT with the smaller effective gap also benefits from much fewer target hits.

## I  Analysis of SVT with individual privacy charging

Our improved SVT with individual charging is described in Algorithm 5 (see Section 2.8).

We establish the privacy guarantee:

*Proof of Theorem 2.9.* We apply simulation-based privacy analysis (see Section A.5). Consider two neighboring datasets $D$ and $D' = D \cup \{x\}$. The only queries where potentially $f(D) \neq f(D')$ and we may need to call the data holder are those with $f(x) \neq 0$. Note that for every $x' \in D$, the counter $C_{x'}$ is the same during the execution of Algorithm 5 on either $D$ or $D'$. This is because the update of $C_{x'}$ depends only on the published results and $f_i(x')$, both of which are public information. Hence, we can think of the processing of $C_{x'}$ as a post-processing when we analyze the privacy property between $D$ and $D'$.

After $x$ is removed, the response on $D$ and $D'$ is the same, and the data holder does not need to be called. Before $x$ is removed from $D'$, we need to consider the queries such that $f(x) \neq 0$ while $C_x < \tau$. Note that this is equivalent to a sequence of `AboveThreshold` tests to linear queries, we apply TCT analysis with `ConditionalRelease` applied with above threshold responses. The claim follows from Theorem B.4. □

We also add that Algorithm 5 can be implemented with `BetweenThresholds` test (see Section 2.6), the extension is straightforward with the respective privacy bounds following from Lemma G.1 ($q$ value for target hit).

## J  Private Selection

In this section we provide proofs and additional details for private selection in TCT (Sections 2.4 and 2.4.1). Let $\mathcal{A}_1, \ldots, \mathcal{A}_m$ be of $m$ private algorithms that return results with quality scores. The private selection task asks us to select the best algorithm from the $m$ candidates. The one-shot selection described in Algorithm 3 (with $k = 1$) runs each algorithm once and returns the response with highest quality.

It is shown in [21] that if each $\mathcal{A}_i$ is $(\varepsilon, 0)$-DP then the one-shot selection algorithm degrades the privacy bound to $(m\varepsilon, 0)$-DP. However, if we relax the requirement to approximate DP, we can show that one-shot selection is $(O(\log(1/\delta)\varepsilon), \delta)$-DP, which is independent of $m$ (the number of candidates). Moreover, in light of a lower-bound example by [21], Theorem J.1 is tight up to constant factors.

Formally, our theorem can be stated as

**Theorem J.1.** *Suppose $\varepsilon < 1$. Let $\mathcal{A}_1, \ldots, \mathcal{A}_m : X^n \to \mathcal{Y} \times \mathbb{R}$ be a list of $(\varepsilon, \delta_i)$-DP algorithms, where the output of $\mathcal{A}_i$ consists of a solution $y \in \mathcal{Y}$ and a score $s \in \mathbb{R}$. Denote by $\mathrm{Best}(\mathcal{A}_1, \ldots, \mathcal{A}_m)$ the following algorithm (Algorithm 3 with $k = 1$): run each $\mathcal{A}_1, \ldots, \mathcal{A}_m$ once, get $m$ results $(y_1, s_1), \ldots, (y_m, s_m)$, and output $(y_{i^*}, s_{i^*})$ where $i^* = \arg\max_i s_i$.*

*Then, for every $\delta \in (0, 1)$, $\mathrm{Best}(\mathcal{A}_1, \ldots, \mathcal{A}_m)$ satisfies $(\varepsilon', \delta')$-DP where $\varepsilon' = O(\varepsilon \log(1/\delta))$, $\delta' = \delta + \sum_i \delta_i$.*

*Proof.* **Discrete scores.** We start by considering the case that the output scores from $\mathcal{A}_1, \ldots, \mathcal{A}_m$ always lie in a *finite* set $X \subseteq \mathbb{R}$. The case with continuous scores can be analyzed by a discretization argument.

Fix $D^0, D^1$ to be a pair of adjacent data sets. We consider the following implementation of the vanilla private selection.

---

**Algorithm 9:** Private Selection: A Simulation

**Input:** Private data set $D$. The set $X$ defined above.
**for** $i = 1, \ldots, m$ **do**
  $(y_i, s_i) \leftarrow \mathcal{A}_i(D)$
**for** $\hat{s} \in X$ *in the decreasing order* **do**
  **for** $i = 1, \ldots, m$ **do**
    **if** $s_i \geq \hat{s}$ **then**
      **return** $(y_i, s_i)$

---

Assuming the score of $\mathcal{A}_i(D)$ always lies in the set $X$, it is easy to see that Algorithm 9 simulates the top-1 one-shot selection algorithm (Algorithm 3 with $k = 1$) perfectly. Namely, Algorithm 9 first runs each $\mathcal{A}_i(D)$ once and collects $m$ results. Then, the algorithm searches for the *lowest* $\hat{s} \in X$ such that there is a pair $(y_i, s_i)$ with a score of at least $s_i \geq \hat{s}$. The algorithm then publishes this score.

On the other hand, we note that Algorithm 9 can be implemented by the conditional release with revisions framework (Algorithm 2). Namely, Algorithm 9 first runs each private algorithm once and stores all the outcomes. Then the algorithm gradually extends the target set (namely, when the algorithm is searching for the threshold $\hat{s}$, the target set is $\{(y, s) : s \geq \hat{s}\}$), and tries to find an outcome in the target. Therefore, it follows from Lemma 2.5 and Theorem B.4 that Algorithm 9 is $(O(\varepsilon \log(1/\delta)), \delta + \sum_i \delta_i)$-DP.

**Continuous scores.** We then consider the case that the distributions of the scores of $\mathcal{A}_1(D), \ldots, \mathcal{A}_K(D)$ are *continuous* over $\mathbb{R}$. We additionally assume that the distribution has no "point mass". This is to say, for every $i \in [m]$ and $\hat{s} \in \mathbb{R}$, it holds that

$$\lim_{\Delta \to 0} \Pr_{(y_i, s_i) \sim \mathcal{A}_i(D)}[\hat{s} - \Delta \leq s \leq \hat{s} + \Delta] = 0.$$

This assumption is without loss of generality because we can always add a tiny perturbation to the original output score of $\mathcal{A}_i(D)$.

Fix $D, D'$ as two neighboring data sets. We show that the vanilla selection algorithm preserves differential privacy between $D$ and $D'$.

Let $\eta > 0$ be an arbitrarily small real number. Set $M = \frac{10 \cdot m^4}{\eta}$. For each $\ell \in [1, M]$, let $q_\ell \in \mathbb{R}$ be the unique real such that

$$\Pr_{i \sim [m], (y_i, s_i) \sim \mathcal{A}_i(D)}[s_i \geq q_\ell] = \frac{\ell}{M + 1}.$$

Similarly we define $q'_\ell$ with respect to $\mathcal{A}_i(D')$. Let $X = \{q_\ell, q'_\ell\}$.

Now, consider running Algorithm 9 with the set $X$ and candidate algorithms $\mathcal{A}_1, \ldots, \mathcal{A}_K$ on $D$ or $D'$. Sort elements of $X$ in the increasing order, which we denote as $X = \{\hat{q}_1 \leq \cdots \leq \hat{q}_m\}$. After

sampling $\mathcal{A}_i(D)$ for each $i \in [m]$, Algorithm 9 fails to return the best outcome only if one of the following events happens.

- The best outcome $(y^*, s^*)$ satisfies that $s^* < \hat{q}_1$.

- There are two outcomes $(y_i, s_i)$ and $(y_j, s_j)$ such that $s_i, s_j \in [\hat{q}_\ell, \hat{q}_{\ell+1})$ for some $\ell \in [n]$.

If Item 1 happens, Algorithm 9 does not output anything. If Item 2 happens, then it might be possible that $i < j, s_i > s_j$, but Algorithm 9 outputs $s_i$.

It is easy to see that Event 1 happens with probability at most $\frac{m^2}{M} \leq \eta$ by the construction of $X$. Event 2 happens with probability at most $M \cdot \frac{m^4}{M^2} \leq \eta$. Therefore, the output distribution of Algorithm 9 differs from the true best outcome by at most $O(\eta)$ in the statistical distance. Taking the limit $\eta \to 0$ completes the proof. $\qquad\square$

**Remark J.2.** *Theorem J.1 shows that there is a factor of $\log(1/\delta)$ overhead when we run top-1 one-shot private selection (Algorithm 9) only once. Nevertheless, we observe that if we compose top-1 one-shot selection with other algorithms under the TCT framework (e.g., compose multiple top-1 one-shot selections, generalized private testing, or any other applications mentioned in this paper)), then* on-average *we only pay $4\varepsilon$ privacy cost (one* NotPrior *target hit with a $2\varepsilon$-DP algorithm) per top-1 selection (assuming $\varepsilon$ is sufficiently small so that $e^\varepsilon \approx 1$). In particular, adaptively performing $c$ executions of top-1 selection is $(\varepsilon', \delta)$-DP where $\varepsilon' = \varepsilon \cdot (4\sqrt{c \log(1/\delta)} + o(\sqrt{c}))$.*

*Liu and Talwar [21] established a lower bound of $2\varepsilon$ on the privacy of a more relaxed top-1 selection task. Hence, there is a factor of 2 gap between this lower bound and our privacy analysis. Note that for the simpler task of one-shot above threshold score (discussed in Section 2.4.1), where the goal is to return a response that is above the threshold if there is one, can be implemented using a single target hit on Conditional Release call (without revise) and this matches the lower bound of $2\varepsilon$. We therefore suspect that it might be possible to tighten the privacy analysis of top-1 one-shot selection. We leave it as an interesting question for followup work.*

### J.1 One-Shot Top-$k$ Selection

In this section, we prove our results for top-$k$ selection.

We consider the natural one-shot algorithm for top-$k$ selection described in Algorithm 3, which (as mentioned in the introduction) generalizes the results presented in [9, 29], which were tailored for selecting from 1-Lipschitz functions, using the Exponential Mechanism or the Report-Noise-Max paradigm.

We prove the following privacy theorem for Algorithm 3.

**Theorem J.3.** *Suppose $\varepsilon < 1$. Assume that each $\mathcal{A}_i$ is $(\varepsilon, 0)$-DP. Then, for every $\delta \in (0, 1)$, Algorithm 3 is $(\varepsilon \cdot O(\sqrt{k \log(\frac{1}{\delta})} + \log(\frac{1}{\delta})), \delta)$-DP.*

**Remark J.4.** *The constant hidden in the big-Oh depends on $\varepsilon$. For the setting that $\varepsilon$ is close to zero so that $e^\varepsilon \approx 1$ and $\delta \geq 2^{o(k)}$, the privacy bound is roughly $(\varepsilon', \delta)$-DP where $\varepsilon' = \varepsilon \cdot (4\sqrt{k \log(1/\delta)} + o(\sqrt{k}))$.*

**Remark J.5.** *We can take $\mathcal{A}_i$ as the Laplace mechanism applied to a 1-Lipschisz quality function $f_i$ (namely, $\mathcal{A}_i(D)$ outputs a pair $(i, f_i(D) + \mathbf{Lap}(1/\varepsilon))$, where $i$ denotes the ID of the $i$-th candidate, and $f_i(D) + \mathbf{Lap}(1/\varepsilon)$ is the noisy quality score of Candidate $i$ with respect to the data $D$). In this way, Theoerem J.3 recovers the main result of [29].*

*Moreover, Theorem J.3 improves over [29] from three aspects: Firstly, Theorem J.3 allows us to report the noisy quality scores of selected candidates for free, while [29] needs to run one additional round of Laplace mechanism to publish the quality scores. Second, our privacy bound has no dependence on $m$, while the bound in the prior work [29] was $(O(\varepsilon\sqrt{k \log(m/\delta)}), \delta)$-DP. Lastly, Theorem J.3 applies more generally to* any *private-preserving algorithms, instead of the classic Laplace mechanism.*

*Proof.* The proof is similar to that of Theorem J.1. Namely, we run each $\mathcal{A}_i(D)$ once and store all results. Then we maintain a threshold $T$, which starts with $T = \infty$. We gradually decrease $T$,

and use Algorithm 2 (Conditional Release with Revised Calls) to find outcomes with a quality score larger than $T$. We keep this process until we identify $k$ largest outcomes. The claimed privacy bound now follows from Lemma 2.5 and Theorem B.4. □

