267 thresholds) and it was shown that the overall privacy costs may only depend on the "between"
268 outcomes. Their analysis required that $t_r - t_l \geq (12/\varepsilon)(\log(10/\varepsilon) + \log(1/\delta) + 1)$. We consider
269 the "natural" private `BetweenThresholds` classifier that compares the value with added $\mathbf{Lap}(1/\varepsilon)$
270 noise to the thresholds. We show (see Section G) that the "between" outcome is a target with
271 $q \geq \left(1 - e^{-(t_r-t_l)\varepsilon}\right) \cdot \frac{1}{e^\varepsilon+1}$. Note that the $q$-value is smaller by a factor of $(1 - e^{-(t_r-t_l)\varepsilon})$ compared
272 with `NotPrior` targets. Therefore, there is smooth degradation in the effectiveness of the between
273 outcome as the target as the gap $t_r - t_l$ decreases, and matching `AboveThreshold` when the gap is
274 large. Also note that we require much smaller gaps $t_r - t_l$ compared with [3], also asymptotically
275 ($O(\log(1/\varepsilon))$ factor improvement). This brings `BetweenThresholds` into the practical regime.

276 We can compare an `AboveThreshold` test with a threshold $t$ with a `BetweenThresholds` classifier
277 with $t_l = t - 1/\varepsilon$ and $t_r = t + 1/\varepsilon$. Surprisingly perhaps, despite `BetweenThresholds` being *more*
278 *informative* than `AboveThreshold`, as it provides more granular information on the value, its privacy
279 cost is *lower* for queries where values are either well above or well below the thresholds (since target
280 hits are unlikely also when queries are well above the threshold). Somehow, the addition of a third

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

** $\quad\quad\quad\quad\quad\quad\quad\quad\quad\quad\quad\quad\quad\quad\quad\quad\quad\quad\quad\quad$ `// Main loop`
$\quad$ **Receive** $(\mathcal{A}_i, \top_i)$ where $\mathcal{A}_i$ is an $(\varepsilon, \delta_i)$-DP mechanism, and $\top_i$ is a $q$-target with $(\varepsilon, \delta_i)$ for $\mathcal{A}$
$\quad$ $r \leftarrow \mathcal{A}_i(D)$
$\quad$ **if** $C_\delta + \delta_i > \tau_\delta$ **then Halt**
$\quad$ $C_\delta \leftarrow C_\delta + \delta$ $\quad\quad\quad\quad\quad\quad\quad\quad\quad\quad\quad\quad\quad\quad\quad\quad$ `// TCT charge for` $\delta_i$
$\quad$ **Publish** $r$
$\quad$ **if** $r \in \top$ **then** $\quad\quad\quad\quad\quad\quad\quad\quad\quad\quad$ `// TCT Charge for a` $q$`-target hit with` $\varepsilon$
$\quad\quad$ $C \leftarrow C + 1$
$\quad\quad$ **if** $C = \tau$ **then Halt**

---

---

**Algorithm 7:** Simulation of Target Charging

---

**Input:** Two neighboring datasets $D^0, D^1$, private $b \in \{0, 1\}$, $\tau \in \mathbb{N}, \tau_\delta \in \mathbb{R}_{\geq 0}, q \in [0, 1], \alpha > 0$.

$C \leftarrow 0, C_\delta \leftarrow 0, h \leftarrow 0$ $\quad$ `// Initialize;` $h$ `is a counter on the number of non-fail calls to data holder`
**for** $i = 1, \ldots$ **do** $\quad\quad\quad\quad\quad\quad\quad\quad\quad\quad\quad\quad\quad\quad\quad\quad\quad\quad\quad\quad$ `// Main loop`
$\quad$ **Receive** $(\mathcal{A}_i, \top_i)$ where $\mathcal{A}_i$ is an $(\varepsilon, \delta_i)$-DP mechanism, and $\top_i$ is a $q$-target with $(\varepsilon, \delta_i)$ for $\mathcal{A}$
$\quad$ **if** $C_\delta + \delta_i > \tau_\delta$ **then Halt**
$\quad$ $C_\delta \leftarrow C_\delta + \delta$
$\quad$ Let $p \in [0, 1]$, $\mathbf{C}, \mathbf{B}^0 \approx_\varepsilon \mathbf{B}^1$, and $\mathbf{E}^b$ (for $b \in \{0, 1\}$) such that
$\quad\quad$ $\mathcal{A}(D^b) \equiv (1 - \delta) \cdot (p \cdot \mathbf{C} + (1 - p) \cdot \mathbf{B}^b) + \delta \cdot \mathbf{E}^b$ $\quad\quad\quad\quad$ `// By Definition B.2`
$\quad$ **if** $\mathbf{Ber}(\delta) \equiv 1$ **then** $\quad\quad\quad\quad\quad\quad\quad\quad$ `// Non-private Data Holder call with Failure`
$\quad\quad$ **Fail**
$\quad\quad$ **Publish** $r \sim \mathbf{E}^b$
$\quad$ **else**
$\quad\quad$ **if** $\mathbf{Ber}(p) \equiv 1$ **then**
$\quad\quad\quad$ **Publish** $r \sim \mathbf{C}$ $\quad\quad\quad\quad\quad\quad\quad\quad\quad\quad$ `// No access to data holder`
$\quad\quad$ **else**
$\quad\quad\quad$ **Publish** $r \sim \mathbf{B}^b$ $\quad\quad\quad\quad\quad\quad\quad\quad\quad\quad$ `//` $\varepsilon$`-DP Data Holder Call`
$\quad\quad\quad$ $h \leftarrow h + 1$ $\quad\quad\quad\quad\quad\quad\quad\quad$ `// counter of` $\varepsilon$`-private data holder calls`
$\quad\quad\quad$ **if** $h > (1 + \alpha)\tau/q$ **then** $\quad\quad\quad\quad\quad$ `// Number of Holder calls exceeded limit`
$\quad\quad\quad\quad$ **Fail**
$\quad\quad\quad$ **if** $r \in \top$ **then** $\quad\quad\quad\quad\quad\quad\quad\quad\quad\quad$ `// outcome is a target hit`
$\quad\quad\quad\quad$ $C \leftarrow C + 1$

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

*Moreover, Theorem I.3 improves over [29] from three aspects: Firstly, Theorem I.3 allows us to report the noisy quality scores of selected candidates for free, while [29] needs to run one additional round of Laplace mechanism to publish the quality scores. Second, our privacy bound has no dependence on $m$, while the bound in the prior work [29] was $(O(\varepsilon\sqrt{k\log(m/\delta)}), \delta)$-DP. Lastly, Theorem I.3 applies more generally to* any *private-preserving algorithms, instead of the classic Laplace mechanism.*

*Proof.* The proof is similar to that of Theorem I.1. Namely, we run each $\mathcal{A}_i(D)$ once and store all results. Then we maintain a threshold $T$, which starts with $T = \infty$. We gradually decrease $T$, and use Algorithm 2 (Conditional Release with Revised Calls) to find outcomes with a quality score larger than $T$. We keep this process until we identify $k$ largest outcomes. The claimed privacy bound now follows from Lemma 2.5 and Theorem B.4. □