# OpenReview forum: "The Target-Charging Technique for Privacy Analysis across Interactive Computations"
_NeurIPS.cc/2023/Conference — NeurIPS 2023 poster_

### Official Review · Reviewer_fHHh · 2023-07-01

**Soundness:** 4 excellent
**Presentation:** 4 excellent
**Contribution:** 3 good
**Rating:** 7
**Confidence:** 3

**Summary:**

Drawing inspiration from the Sparse Vector Technique (SVT), the authors introduce a privacy analysis framework called Target Charging Technique (TCT). TCT operates by primarily accounting for the privacy loss incurred through queries that hit pre-specified target sets. This framework relies on the concept of q-targets, which model a structural property of the distribution of privacy mechanism outputs. The authors show that q-targets appear in a variety of use cases, leading to natural applications of TCT. Of particular interest is the application of TCT in the privacy analysis of one-shot k-top selection and conditional releases. Also, the authors theoretically demonstrate that the parameter degradation of TCT improves over known techniques.

**Strengths:**

1. The problem is rather natural, as well as the proposed algorithm

2. The framework is flexible enough to accommodate interesting applications, e.g., one-shot k-top selection and conditional releases. Furthermore, it can be generalized to multiple targets.

3. The technical part of the paper seems to be solid.

**Weaknesses:**

The reviewer acknowledges that the primary focus of the present paper is to establish the foundations of TCT and prove some of its theoretical properties. However, the absence of numerical experiments poses a significant weakness. Considering the authors' expressed intention for TCT to be adopted in practice, it is important to supplement the theoretical analysis with concrete evidence through the inclusion of some simulations. Such simulations would provide valuable insights into the practical implications and potential benefits of employing TCT in real-world scenarios.

**Questions:**

As highlighted in the Weaknesses Section, the primary limitation of the present paper lies in the absence of numerical experiments that demonstrate the benefits of the proposed framework. It would be advantageous for the reader if the authors could include some illustrative simulations, even if they are relatively simple in nature. This addition would provide valuable insights into the practical advantages of the proposed framework.

**Limitations:**

The authors' treatment of the limitations of the proposed method could be improved. Specifically, there is a lack of empirical experimentation to showcase the benefits of TCT.

---

> ### Author Rebuttal · Authors · 2023-08-10
>
> Thank you for your review!
>
> Response to “weaknesses”:  Please see our general response with numerical comparisons and a demonstration of the practicality of TCT and concrete improvements over prior work.  We hope this will at least partially address the concern!  We plan to incorporate this in the paper.

---

> > ### Comment · Reviewer_fHHh · 2023-08-16
> >
> > Thanks a lot for adding a numerical comparison against prior work. I slightly raised my score.

---

> > > ### Author Response · Authors · 2023-08-18
> > >
> > > Thank you!

---

### Official Review · Reviewer_o37L · 2023-07-06

**Soundness:** 4 excellent
**Presentation:** 3 good
**Contribution:** 3 good
**Rating:** 7
**Confidence:** 2

**Summary:**

This paper introduces the Target Charging Technique (TCT) privacy analysis framework that focuses on providing better privacy utility trade-off on sensitive dataset with multiple differential private algorithm  access. The essential idea is to only pay budget cost to the positive target and negative targets become free. TCT framework can improve several classical DP algorithms such as private Top-K, conditional release, private learning with non-private models etc.

**Strengths:**

This paper studies a very important problem and proposes a very generalized framework that works for several classical algorithms and with many applications. The framework provide tighter privacy-utility trade off by considering the amortized overheads over user desired target. When the number of hits is large, the $log(1/\delta)$ privacy charge amortized to O(1) per target hit.

The intuition behind this paper is well presented in which if the computations do not touch user targets then the overhead should be small.



**Weaknesses:**

The assumption in q-Target definition seems to be stringent. How to ensure the randomized algorithm M satisfy such condition?

While it is a formal paper, it would still benefits with some empirical evidence.

**Questions:**

See weakness.

---

> ### Author Rebuttal · Authors · 2023-08-10
>
> Thank you for your review!
>
> Response to “weaknesses”:
>
> 1. [q-target] We agree that the definition is formal. We do not ensure that this holds (there is always a target with $q=1$ but this is not very interesting) but instead show how to use its existence in privacy analysis.  We then demonstrate that there are natural q-targets, in particular “not prior” (all outcomes but a selected one in any private algorithm).  Not-prior is our workhorse in multiple applications but we also demonstrate there is a value in this generality (rather than limiting ourselves to “not prior” targets).  For example, in the “BetweenThreshold” application we demonstrate the  $q$-value of the “between” outcome (the target) depends on the width.
>
> 2. [empirical evidence] Please see our general response with numerical comparisons and a demonstration of the practicality of TCT and concrete improvements over prior work.  We plan to incorporate it in the paper.

---

### Official Review · Reviewer_5m6N · 2023-07-07

**Soundness:** 4 excellent
**Presentation:** 4 excellent
**Contribution:** 4 excellent
**Rating:** 8
**Confidence:** 1

**Summary:**

Introduced Target-Charging Technique (TCT) for privacy analysis over interactive private computations

**Strengths:**

1.  Introduced Target-Charging Technique (TCT) for privacy analysis over interactive private computations

**Weaknesses:**

1. They could add experiment part

**Questions:**

1. Thank you author because the best presentation as far as I have seen. Even after this, I will also use comment feature for algorithm or coding snippet of my paper.

**Limitations:**

1. I am missing experimental result.

---

> ### Author Rebuttal · Authors · 2023-08-10
>
> Thank you for your review!
>
> Response to “weaknesses”:  Please see our general response with numerical comparisons and a demonstration of the practicality of TCT and concrete improvements over prior work.

---

### Official Review · Reviewer_MAFe · 2023-07-10

**Soundness:** 3 good
**Presentation:** 2 fair
**Contribution:** 3 good
**Rating:** 7
**Confidence:** 4

**Summary:**

This paper proposes Target Charging Techniques: a method that generalizes sparse vector technique and top-k selection from private candidates. Specifically their goal is to obtain better privacy guarantees in the settings where only a small number of computations are *successful*. One of their main ideas is to consider a prior on the output that has high probability (some sort of a default output), and only pay in privacy if the output of the algorithm is not equal to that prior.

They suggest a definition "$q$-target" for a subset of events $T$ and a private algorithm $M$. Roughly speaking, this definition helps us measure the actual privacy exposure when the algorithm $M$ hits an event in T. For every $1/q$ private accesses we're going to have $1$ hit (we'll have $ M(D)$ in $T$).

Suppose we have a dataset $D$, a set of $\epsilon$-DP algorithms $A_i$'s and events $T_i$, and a threshold $\tau$. They show an algorithm that outputs $A_i(D)$'s until $\tau$ many of $A_i(D)$'s are in $T_i$ and then stops is $(\tau / q \epsilon, \exp(O(\tau))$ private.

In the rest of the paper they apply their technique to multiple use cases.

Some examples of the use cases of this technique include: private learning when we want to release only if the answer has high quality, greedy coverage, and above threshold tests.

**Strengths:**

This paper provides a simple framework for accounting for privacy that is applicable to a number of problems in dp such as top-k selection and sparse vector technique with individual privacy charging. These problems mostly had tailor-made algorithms and analysis prior this work. The framework in this paper recovers some of the previously known bounds on these problems and improves on some of them, often through a simpler analysis.

**Weaknesses:**

I expected the authors to include at least one setting where this framework significantly improves upon prior work on a specific application. This paper provides a pipeline that is simple, nice, and will probably be useful. Still, I am not convinced that the applications of this framework provide a significant improvement in terms of results compared to prior work.

**Questions:**

suggestion: add some text around definition 2.1 to explain what's going on in the definition and to help interpret lemma 2.2

Lemma 2.2 talks about $A_i$'s and $\tau_i$'s, but algorithm 1 only has $A$ and $\tau$.

In line 374: can you give a more complete explanation of what their result is and what are the guarantees that you obtain?

**Limitations:**

The authors have adequately addressed the limitations.

---

> ### Author Rebuttal · Authors · 2023-08-10
>
> Thank you for your review!
>
> Response to “weaknesses”:  Please see our general response regarding numerical comparisons and a demonstration of the practicality of TCT and concrete improvements over prior work.
>
> Response to Questions:
>
>   -- Definition 2.1 formalizes the notion of a “target” that was discussed in the introduction.  We will add a backward reference to the informal use in the intro.
>
>  -- We modified the statements of Lemma 2.2  to remove the explicit index notation. It is the case that new algorithm-target pairs $(A_i,\tau_i)$ are input in each iteration but we made the notation consistent.
>
> -- Line 374: please see additional details we provide on “SVT with individual privacy charging” in the general response.  The idea in [KMS] [20]  is to do a more fine-grained privacy analysis of threshold tests on counting queries that keeps a budget for each item (that is charged only when the item “participates”) instead of one overall budget.  This allows us to answer more queries with the same privacy budget.  Compared to the algorithm (and analysis) of [20], our Algorithm 5 with TCT analysis is simpler (adds noise once), more informative within the privacy budget as it can also release actual estimates with above-threshold responses, and is vastly more accurate:  can add much smaller noise for the same privacy budget (see general response).

---

> > ### Comment · Reviewer_MAFe · 2023-08-19
> >
> > I thank the authors for their response. After having read the review of other reviewers and the authors' rebuttal I have updated my review. I think the improvement over Kaplan-Mansour-Stemmer is interesting. I suggest that the authors present these improvements more clearly in the paper.

---

> > > ### Author Response · Authors · 2023-08-20
> > >
> > > Thank you for your review and comments! We will incorporate your suggestions in our revision.

---

### Official Review · Reviewer_xrUp · 2023-07-21

**Soundness:** 3 good
**Presentation:** 3 good
**Contribution:** 3 good
**Rating:** 6
**Confidence:** 2

**Summary:**

This paper presents a new privacy analysis framework called Target Charging Technique (TCT). Designed for interactive scenarios where sensitive datasets are frequently accessed using differentially private algorithms, TCT offers a unique perspective. Unlike traditional composition schemes where privacy assurances degrade with multiple data accesses, TCT provides an alternative approach. It allows computations that avoid hitting a specified target to occur without significant cost but imposes a minor overhead on computations that do meet their targets. Furthermore, TCT generalizes tools such as the sparse vector technique and top-k selection from private candidates and extends their remarkable privacy enhancement benefits from noisy Lipschitz functions to general private algorithms.


**Strengths:**

1. This paper is well written. The details of movtivation, methodology, proofs are described clearly.
2. The insight of this paper is quite reasonable and the proposed approach looks novel.

**Weaknesses:**

1. The related work section is missing.
2. The effectiveness of this approach is not clear to me. What are the main improvements offered by this method?

**Questions:**

It seems that this work is very promising in theory. Have you tested it on some real-world scenarios? Will it become open-source?

**Limitations:**

See weakness.

---

> ### Author Rebuttal · Authors · 2023-08-10
>
> Thank you for your review!
>
> Response to Question: Please see our general response regarding numerical comparisons and a demonstration of the practicality of TCT and improvement over prior work.  As for open-source: TCT privacy analysis is simple to add and we hope it will be  incorporated in DP libraries.
>
> Response to “Weaknesses”:
>
> 1. Our work is related to vast literature on SVT and selection and we included many citations and discussions of related work throughout the introduction. The prior work on SVT that can be viewed as a predecessor of Target Charging is the paper by Hardt and Rothblum (FOCS 2010) [18].  All that said, we are very happy to include a related work section if this is the preferred format.
>
> 2. Please see our general response where we numerically demonstrate the improvements over prior work.

---

> > ### Comment · Reviewer_xrUp · 2023-08-17
> > **Reply to Authors**
> >
> > Thanks for your rebuttal. My concerns have been addressed and I raise my rating to 6.

---

> > > ### Author Response · Authors · 2023-08-18
> > >
> > > Thank you!

---

### Official Review · Reviewer_ot2S · 2023-07-25

**Soundness:** 3 good
**Presentation:** 2 fair
**Contribution:** 2 fair
**Rating:** 6
**Confidence:** 2

**Summary:**

The submission studies a privacy analysis framework for interactive computations. The main contribution is extending the conventional differential privacy analysis framework, namely the sparse vector technique, which counts the privacy cost only for the positive response. Particularly, the proposed framework, namely the target charging technique, generalizes from the specific constraints (e.g., threshold) of positive response to general expressions that are not necessarily threshold. The overall analyses are regarding the $(\epsilon,\delta)$-DP.

**Strengths:**

The paper is well-organized. I think the problem of privacy analysis for interactive computing is an essential topic and the extension/generalization made in this work is significant.
I think that the privacy analysis shown in Lemma 2.2 for a general system is important in DP literature.
The list of application scenarios of the proposed framework is well-categorized and shows the applicability of the proposed technique in practice.

**Weaknesses:**

I think the results and definitions are hard to read. In particular, it is hard to see how one can distinguish the q value of a q-target. It is also not straightforward to understand the notPrior target.
I know this paper is a theory paper, but it would be good to show some experimental results of the application of the main results, which makes the paper more sound.

**Questions:**

1) As written in the weaknesses part, I cannot see how the q value is attained from the target setting in practice. Could you provide a (toy) example for the q-target?

2) What happens if the $\epsilon$ is not too small (e.g. $\epsilon=10$ in Lemma 2.3)? It seems the resulting $\epsilon$ would be very large, which is not acceptable in general. Is there any way to compensate for it?

---

> ### Author Rebuttal · Authors · 2023-08-10
>
> Thank you for your review!
>
> Answers:
> As for experimental results, see our general response!
>
> Question 1:  $q$-targets:  We list examples of useful q-targets.  For private testing, the $q$-target can be (selected one of) true/false responses.  For Top-$k$ selection, the $q$-targets are mapped to being in the top-$k$.  For “between thresholds” tests,  the $q$-target is simply a between outcome.  The only place where the $q$- target is “manufactured” is in our “boundary wrapper” method.
> $q$-values are derived for each of the cases.  Nearly all of our use cases directly map to the “not prior” q-value that (for small $\varepsilon$) is close to $1/2$.  The “between thresholds” application includes an exact calculation of $q$ as a function of the  “gap” between the thresholds.
>
> Question 2:  Note that the value of $\varepsilon$ in the context of Lemma 2.3 is that of a single call to a DP algorithm on our dataset.  This value is typically very small $\varepsilon \ll 1$ and therefore $q\approx ½$.   In the TCT setting, we assume that there are multiple applications of private algorithms to our dataset, and tens (or hundreds) of them are “target hits.”  The overall privacy cost is a composition over target hits.  We can set out overall privacy budget to have $\varepsilon=1$ or if one wishes $\varepsilon=10$.  But regardless, the privacy parameter values of each call are small.

---

> > ### Comment · Reviewer_ot2S · 2023-08-16
> >
> > Thank you for the reply and the rebuttal. Those help me have a better understanding of the contribution.
> > I raise my score.

---

> > > ### Author Response · Authors · 2023-08-18
> > >
> > > Thank you!

---

### Author Rebuttal · Authors · 2023-08-10

We are grateful to the reviewers for their comments.  All reviewers expressed a desire to see some numerical comparisons of the gains of TCT with respect to prior work. In this comment, we place our results in this context and plan to revise our presentation accordingly.  Responses to additional comments are provided individually.

– “BetweenThresholds” queries (see Section 2.6):

For dataset $D = \{x_1,\dots, x_n\}$ and threshold $0\leq t_l<t_r\leq 1$, we process a sequence of queries specified by functions $f$ from the domain to $[0,1]$.  The aim is to compare $\frac{1}{n} \sum_{i=1}^{n} f(x_i)$ to the thresholds and return "below","between", and "above".  We include a plot that demonstrates significant improvement over prior work. The plot reports the number of "target hit" queries ($y$-axis) that we can perform with an overall privacy budget of $\varepsilon=1$ subject to accuracy requirement of relative error $0.01$ with confidence $1-1/n$.  This as a function of the dataset size $n$ ($x$-axis). "Target hits" are “between” responses. Note a factor of 95 improvement of TCT over the method of Bun-Steinke-Ullman [3]. We also performed an “optimistic” analysis of the constants in BSU [3], that can be interpreted as an “upper bound” on what is achievable with their approach. TCT gained a factor of 6 improvement even compared with that optimistic bound.

TCT offers additional advantages over [3]:  Our algorithm is simpler and natural (add Laplace noise and compare with thresholds).  Importantly, it allows for asymptotically narrower gaps between the thresholds compared with [3]:  [3] requires a gap of
$|t_r -t_l | >= \frac{12(\log(1/\varepsilon) + \log(1/\delta))}{\varepsilon}$, whereas TCT allows for gaps as small as $2/\varepsilon$ with almost no decrease in the $q$-value. This is particularly significant in applications where we are interested in a single threshold and incurring privacy loss only on queries that are close to that threshold.  The narrower gap means much fewer target hits.

– SVT with  Individual Privacy Charging (Section 2.8 for details):

We compare TCT with the prior work of Kaplan Mansour Stemmer [20]. We can achieve the same privacy guarantees with much lower levels of  noise:  [20] uses standard deviation $O( \sqrt{log(1/\delta)} \log(1/\varepsilon) /\varepsilon)$ whereas TCT uses $O(1/\varepsilon)$.  We can not do an empirical comparison because [20] only provided asymptotic analysis with “constants” that appear much larger (we have the same constants as without individual charging).  We can do a forgiving (to [20]) comparison by ignoring constants: If we take $\varepsilon=0.01$ and $\delta  = 10^{-8}$ the noise magnitude  with TCT is smaller by a factor of 20.

Additionally, our algorithm is simpler and the privacy analysis is few lines compared with several pages.  Moreover, TCT analysis allows for releasing estimated counts with no additional privacy charge.



– General gains of TCT

The two examples (BetweenThresholds and Individual charging) are in the “noisy Lipschitz” setting where there is prior work to compare with. Zooming out a bit, TCT is a broad and flexible framework.  Its primary advantage is that it allows for incurring privacy loss only on “target hits” in natural settings that are outside “noisy Lipschitz” regime.  These include one-shot top-k selections and private tests  with general private algorithms (aka not necessarily noisy Lipschitz).  The relevant prior work here is either changing the algorithm or applying DP composition over $n$.  TCT gains are significant as $k$ can be much much smaller than $n$ (the total number of tests or the total number we select from).  Moreover, our analysis does not hide large constants.  Lemma 2.2 states simplified privacy bounds for TCT with some asymptotic notation, but In the supplementary we included exact expressions in the statement of Theorem B.4. We include numerical examples in Remark B.5: TCT analysis generally gains when $n/k > 2.5$ when there are hundreds of target hits. The needed ratio is larger with fewer target hits. For 10 target hits we need $n/k > 12$ to gain over composition.  These regimes are in the practical realm.  In applications such as PATE (see section 2.7.1) one aims to incur privacy loss on a small fraction of examples.

---

### Decision · Program_Chairs · 2023-09-21

**Decision:**

Accept (poster)

**Comment:**

The reviewers were all unanimously supportive of this paper and its contributions. The reviewers felt that the authors addressed any remaining concerns in the discussion period.